# PRDM9 drives the location and rapid evolution of recombination hotspots in salmonid fish

**Marie Raynaud**[1☯]*, **Paola Sanna**[2☯], **Julien Joseph**[3], **Julie Clément**[4], **Yukiko Imai**[5], **Jean-Jacques Lareyre**[6], **Audrey Laurent**[6], **Nicolas Galtier**[1], **Frédéric Baudat**[2], **Laurent Duret**[3‡]*, **Pierre-Alexandre Gagnaire**[1‡]*, **Bernard de Massy**[2‡]*

1 ISEM, Univ Montpellier, CNRS, IRD, Montpellier, France, 2 Institut de Génétique Humaine, Univ Montpellier, Centre National de la Recherche Scientifique, Montpellier, France, 3 Laboratoire de Biométrie et Biologie Évolutive, Universite Claude Bernard Lyon 1, CNRS UMR 5558, Villeurbanne, France, 4 IHPE, Univ Montpellier, CNRS, IFREMER, Univ Perpignan Via Domitia, Perpignan, France, 5 Department of Gene Function and Phenomics, National Institute of Genetics, Mishima, Japan, 6 INRAE, UR1037 Fish Physiology and Genomics, F-35000 Rennes, France

☯ These authors contributed equally to this work.
‡ LD, PAG, and BdM also contributed equally to this work.
* marieraynaud18@hotmail.fr (MR); Laurent.Duret@univ-lyon1.fr (LD); pierre-alexandre. gagnaire@umontpellier.fr (P-AG); bernard.de-massy@igh.cnrs.fr (BdM)

**Data Availability Statement:** Sequencing data from ChIP-seq and called peaks have been deposited in the Gene Expression Omnibus (GEO)

## Abstract

In many eukaryotes, meiotic recombination occurs preferentially at discrete sites, called recombination hotspots. In various lineages, recombination hotspots are located in regions with promoter-like features and are evolutionarily stable. Conversely, in some mammals, hotspots are driven by PRDM9 that targets recombination away from promoters. Paradoxically, PRDM9 induces the self-destruction of its targets and this triggers an ultra-fast evolution of mammalian hotspots. PRDM9 is ancestral to all animals, suggesting a critical importance for the meiotic program, but has been lost in many lineages with surprisingly little effect on meiosis success. However, it is unclear whether the function of PRDM9 described in mammals is shared by other species. To investigate this, we analyzed the recombination landscape of several salmonids, the genome of which harbors one full-length PRDM9 and several truncated paralogs. We identified recombination initiation sites in *Oncorhynchus mykiss* by mapping meiotic DNA double-strand breaks (DSBs). We found that DSBs clustered at hotspots positioned away from promoters, enriched for the H3K4me3 and H3K36me3 and the location of which depended on the genotype of full-length *Prdm9*. We observed a high level of polymorphism in the zinc finger domain of full-length *Prdm9*, indicating diversification driven by positive selection. Moreover, population-scaled recombination maps in *O. mykiss*, *Oncorhynchus kisutch* and *Salmo salar* revealed a rapid turnover of recombination hotspots caused by PRDM9 target motif erosion. Our results imply that PRDM9 function is conserved across vertebrates and that the peculiar evolutionary runaway caused by PRDM9 has been active for several hundred million years.

under accession GSE277449, as part of BioProject PRJNA1162462. Bioinformatic scripts as well as processed data sets (VCF files, LD-based recombination maps and hotspots, PRDM9 protein sequences and multiple alignments, ChIP-Seq fragments and peaks coordinates) are available on Zenodo (https://doi.org/10.5281/zenodo. 11083953). Scripts for ChIP-seq data analysis are available at https://zenodo.org/records/14198856 (SSDS pipeline), https://zenodo.org/records/ 14198863 (SSDS-extra pipeline) and https:// zenodo.org/records/3966161 (Next-flow ChIPseq pipeline).

**Funding:** This project was funded by CNRS (Centre national de la recherche scientifique; https://www. cnrs.fr) and by ANR (Agence nationale de la recherche; https://anr.fr) (HotRec ANR-19-CE12- 0019) to LD, NG and BdM. The funders had no role in study design, data collection and analysis, decision to publish, or preparation of the manuscript.

**Competing interests:** The authors have declared that no competing interests exist.

**Abbreviations:** BS, Barents Sea; ChIP, chromatin immunoprecipitation; CO, crossover; DSB, double-strand break; GD, gene duplication; GP, Gaspesie Peninsula; IDR, irreproducible discovery rate; LD, linkage disequilibrium; MAC, minor allele count; NS, North Sea; SD, segmental duplication; SNP, single-nucleotide polymorphism; SRA, Sequence Read Archive; TE, transposable element; TES, transcription end site; TSS, transcription start site; WGD, whole genome duplication; ZF, zinc finger.

## Introduction

Meiotic recombination (i.e., the exchange of genetic material between homologous chromosomes during meiosis) is highly conserved in a wide range of sexually reproducing eukaryotes, including plants, fungi, and animals [1]. This process is initiated by the programmed formation of DNA double-strand breaks (DSBs), followed by their repair using the homologous chromosome as template. Recombination events can lead to the reciprocal exchange of flanking regions (crossovers, COs) or proceed without reciprocal exchange (non-crossovers, NCOs). COs are essential for the proper segregation of homologous chromosomes [2]. Failure to form COs can lead to aneuploid reproductive cells or to defects in meiotic progression and sterility [3]. Meiotic recombination also plays an important evolutionary role. It increases genetic diversity by creating novel allele combinations [4,5] that facilitate adaptation and the removal of deleterious mutations from natural populations [6–8].

Intriguingly, the CO rate varies not only among species, populations, sexes, and individuals, but also along the genome [9–11]. Broad-scale patterns of variation within chromosomes (at the megabase scale) have been observed in some species: low recombination rate near centromeres and high recombination rate in telomere-proximal regions [12]. At a finer scale (kilobases), CO rate across the genome ranges from nearly uniform (e.g., flies, worms, and honeybees) [13–15] to highly heterogeneous (e.g., yeast, plants, and vertebrates). In such non-uniform recombination landscapes, most recombination events are typically concentrated within short intervals of about 2 kb, called recombination hotspots [16,17]. Studies on the evolutionary dynamics of recombination hotspots have identified 2 alternative mechanisms for controlling hotspot localization. In many eukaryotes (e.g., *Arabidopsis*, budding yeast, swordtail fish, birds, and canids), hotspots tend to be located near chromatin accessible regions enriched for H3K4me3, including promoters and transcription start sites (TSSs) [18–25]. Elevated recombination rates are also observed at transcription end sites (TESs) in plants and birds [25,26]. In dogs and birds, recombination hotspots are particularly associated with TSSs that are located within CpG islands (CGIs) [18,25,27]. Hotspot location is conserved over large evolutionary timescales in birds and yeasts [22,23,25], likely because promoters are evolutionarily stable. However, the generality of this conclusion remains to be evaluated [28]. On the other hand, mammalian species, including primates, mice, and cattle, show a drastically different pattern. Their recombination hotspots tend to occur independently of open chromatin regions [29–32], and their positions evolve rapidly between closely related species and even populations [29,30,33–35]. The genomic location of mammalian hotspots is controlled by the PRDM9 protein [32,36,37] that has 4 canonical domains (KRAB, SSXRD, PR/SET, and zinc finger, ZF), among which the C2H2 ZF domain binds to a specific DNA motif. After PRDM9 binding to this motif, PRDM9 trimethylates H3K4 and H3K36 on adjacent nucleosomes through its SET domain. Then, the proteins required for DSB formation are recruited at PRDM9 binding sites. The formed DSBs are repaired by homologous recombination, leading to COs and NCOs [38]. Two striking evolutionary properties of PRDM9 have been identified. First, PRDM9 triggers the erosion of its binding sites, due to biased gene conversion during DSB repair [32,39,40]. Second, its ZF array presents a very high diversity [41–46] resulting from rapid evolution driven by a Red Queen dynamic in which positive selection favors the formation of new ZF arrays that recognize new binding motifs [39,40,47–50]. This is the direct consequence of PRDM9 binding site erosion that decreases the efficiency of inter-homolog DSB repair, thus leading to lower fitness [47,50–53]. As a result, in *Mus musculus*, strains carrying different PRDM9 alleles generally share only 1% to 3% of DSB hotspots [34], and hotspot locations hardly overlap between humans and chimpanzees [29]. Thus, PRDM9-dependent and -independent hotspots display different genomic locations and also evolutionary lifespan.

This raises the question of why and how the genetically unstable mechanism of PRDM9-directed recombination has evolved [19,54].

Understanding the function and evolutionary dynamics of PRDM9 in mammals has been a major breakthrough [29,32,36,42,55]. Phylogenetic studies of the *Prdm9* gene have revealed the presence of a full-length copy in many metazoans and also repeated partial or complete losses [19,27,54,56]. This is surprising for a gene that controls such a crucial mechanism as reported in mammals. Among vertebrates, fine-scale recombination maps from species lacking *Prdm9* (e.g., birds and dogs) or harboring a truncated KRAB-less *Prdm9* (e.g., swordtail fish or three-spined stickleback), revealed that their recombination hotspots are enriched at CGI-associated promoters [18,19,22,25,27,57], as observed in *Prdm9* knockout mice or rats [30,58]. In snakes, which carry a full-length *Prdm9* copy, the predicted binding sites of PRDM9 alleles are associated with increased recombination rates, which suggests that the sites of recombination are at least in part specified by the DNA binding property of PRDM9 similarly to mammals [59]. Whether PRDM9-mediated epigenetic modifications are functional in snakes is not known. However, snake genomes also show an enrichment of recombination at promoter-like features [59,60] that appears to be Prdm9 independent [59]. Interestingly, all vertebrate species with a full-length PRDM9 show evidence of rapid evolution in its DNA-binding domain, as predicted by the Red Queen model [19]. Furthermore, ZCWPW1, which binds H3K4me3 and H3K36me3, appears to co-evolve with PRDM9 in vertebrates [56]. All these observations suggest that the function of PRDM9, as described in mammals, might be ancestral to all vertebrates, and that the partial or complete loss of *Prdm9* leads to a reversal of the default mechanism of hotspot location at gene promoters. However, it should be noted that with the exception of mammals, current knowledge of PRDM9 function relies only on indirect evidence. Furthermore, with the exception of mammals and snakes, fine-scale recombination landscapes have only been studied in animals lacking a functional PRDM9 (e.g., fruit flies [13], birds [22,25], three-spined stickleback [57,61], swordtail fish [19], lizards [62], and honeybees [15]). Thus, the question of PRDM9 function and how it evolved, particularly whether it was ancestrally involved in regulating recombination hotspots, or whether this function appeared more recently remains to be explored. To address this question, we need to characterize the recombination landscapes in other nonmammalian taxa that harbor PRDM9 and determine whether their characteristics and dynamics are similar to those described in mammals.

To this aim, we investigated the putative function of PRDM9 in salmonids, a diverse family of teleost fishes in which a full-length *Prdm9* has been found [19,56]. Genes that have been shown to co-evolve with *Prdm9* (*Zcwpw1*, *Zcwpw2*, *Tex15*, and *Fbxo47*) are all present in salmonids [56]. Thus, the phylogenetic position of salmonids is ideal for testing the hypothesis of an ancestral PRDM9 role in regulating meiotic recombination in vertebrates. We used the large amount of genomic resources available in salmonids and also generated new data to test the role of PRDM9 in driving the location of recombination events in salmonids. Specifically, if the role of PRDM9 in salmonids were the same as in mammals, we would expect (i) the presence of recombination hotspots; (ii) located away from promoters; (iii) overlapping with enrichment for H3K4me3 and H3K36me3; (iv) showing rapidly evolving landscapes between closely related species and populations; and (v) associated with high diversity of the PRDM9 ZF domain. Importantly, salmonids have undergone 2 rounds of whole genome duplication (WGD) [63–65], offering the opportunity to investigate the impact of gene duplication (GD) on *Prdm9* evolutionary dynamics.

To test these hypotheses, we first analyzed the functional conservation of the many *Prdm9* duplicated copies across the phylogeny of salmonids. We then characterized the functional *Prdm9* allelic diversity in Atlantic salmon and rainbow trout to assess the evolutionary dynamics of the ZF array. We also determined the meiotic DSB landscape in rainbow trout using

chromatin immunoprecipitation (ChIP) of the recombinase DMC1 followed by sequencing, and compared it with the genomic landscapes of the H3K4me3 and H3K36me3 modifications. Lastly, we reconstructed linkage disequilibrium (LD)-based recombination landscapes in 5 populations from 3 different salmonid species to identify hotspots, test their association with genomic features, and measure their evolutionary stability. Our results provide a body of evidence supporting PRDM9 role as a determinant of recombination hotspots in salmonids.

## Results

### Duplication history and differential retention of *Prdm9* paralogs in salmonids

The analysis of the genomes of 12 salmonid species and of northern pike and sea bass (used as outgroups) revealed multiple paralogous copies of the *Prdm9* gene. These paralogs partly resulted from 2 rounds of WGD: the teleost-specific WGD that occurred approximately 320 Mya (referred to as Ts3R) [63,65] and a more recent WGD in the common ancestor of salmonids at approximately 90 Mya, after their speciation with pikes (referred to as Ss4R) [64]. Taking advantage of the known pairs of ohnologous chromosomes resulting from WGD in salmonids [66–68], we reconstructed the duplication history of *Prdm9* paralogs by combining chromosome location information and phylogenetic inference. The number of *Prdm9* paralogs detected per genome ranged from 6 copies in rainbow trout (*Oncorhynchus mykiss*), huchen (*Hucho hucho*), and European grayling (*Thymallus thymallus*), to 14 in lake whitefish (*Coregonus clupeaformis*). Conversely, we found only 3 copies in northern pike (*Esox lucius*). These paralogs clustered into 2 main groups that were previously identified as *Prdm9α* and *Prdm9β* and originated from the Ts3R WGD [19]. We found 2 additional subgroups among the *Prdm9β* copies (referred to as *β1* and *β2*) that were conserved in the 12 salmonid species, but only 1 *β* copy in the outgroups (S1 Fig). The *β* paralogs contained a complete SET domain (but with mutations at the catalytic tyrosine residues) and a conserved ZF domain, but all lacked the KRAB and SSXRD domains, as previously described [19] (S1 Fig). The *α* sequences clustered into 2 well-supported groups of paralogs (named *α1* and *α2*) that could be subdivided in 2 groups of duplicated copies (designated as *α1.1*/*α1.2* and *α2.1*/*α2.2*; Fig 1A). We found the sequence pairs *β1*/*β2*, *α1.1*/*α1.2*, and *α2.1*/*α2.2* in 3 Ss4R ohnologous pairs, suggesting that they originated from the salmonid-specific WGD. We observed an additional subdivision within the *α1* group, with pairs of copies duplicated in tandem present in each pair of ohnologs (i.e., *α1.1 a* and *b* and *α1.2 a* and *b*; Fig 1A and S1 Table). These duplicated copies are found in almost all species, often having the same orientation (S2 Fig). Although no phylogenetic signal was associated with the *a* and *b* copies, probably due to ectopic recombination and gene conversion, these copies are likely to represent a segmental duplication (SD) that preceded the Ss4R WGD. Thus, at least 2 *Prdm9* duplication events (i.e., one leading to *α1*/*α2* and the other to *α1.a*/*α1.b* copies) occurred in addition to the WGD-linked duplications. To summarize, our results indicate that *Prdm9α* and *β* copies originated from the Ts3R WGD. After the divergence of the Esociformes (pike) and Salmoniformes lineages approximately 115 Myrs ago, the *α* copy was duplicated on another chromosome, generating *α1* and *α2* copies. The *α1* copy was subsequently duplicated in tandem, producing *α1.a* and *α1.b* copies on the same chromosome. Lastly, all these copies were duplicated on ohnologous chromosome pairs following the Ss4R WGD. This consensus evolutionary history was accompanied by gene conversion events and lineage-specific duplications and losses that were not fully identified in our analysis (Fig 1B). Most of these gene copies only contained a subset of the 10 expected exons and/or showed signatures of pseudogenization (stop codons, frameshifts), but we also identified some complete *Prdm9* genes, encoding the 4 canonical domains, with conserved catalytic tyrosines in

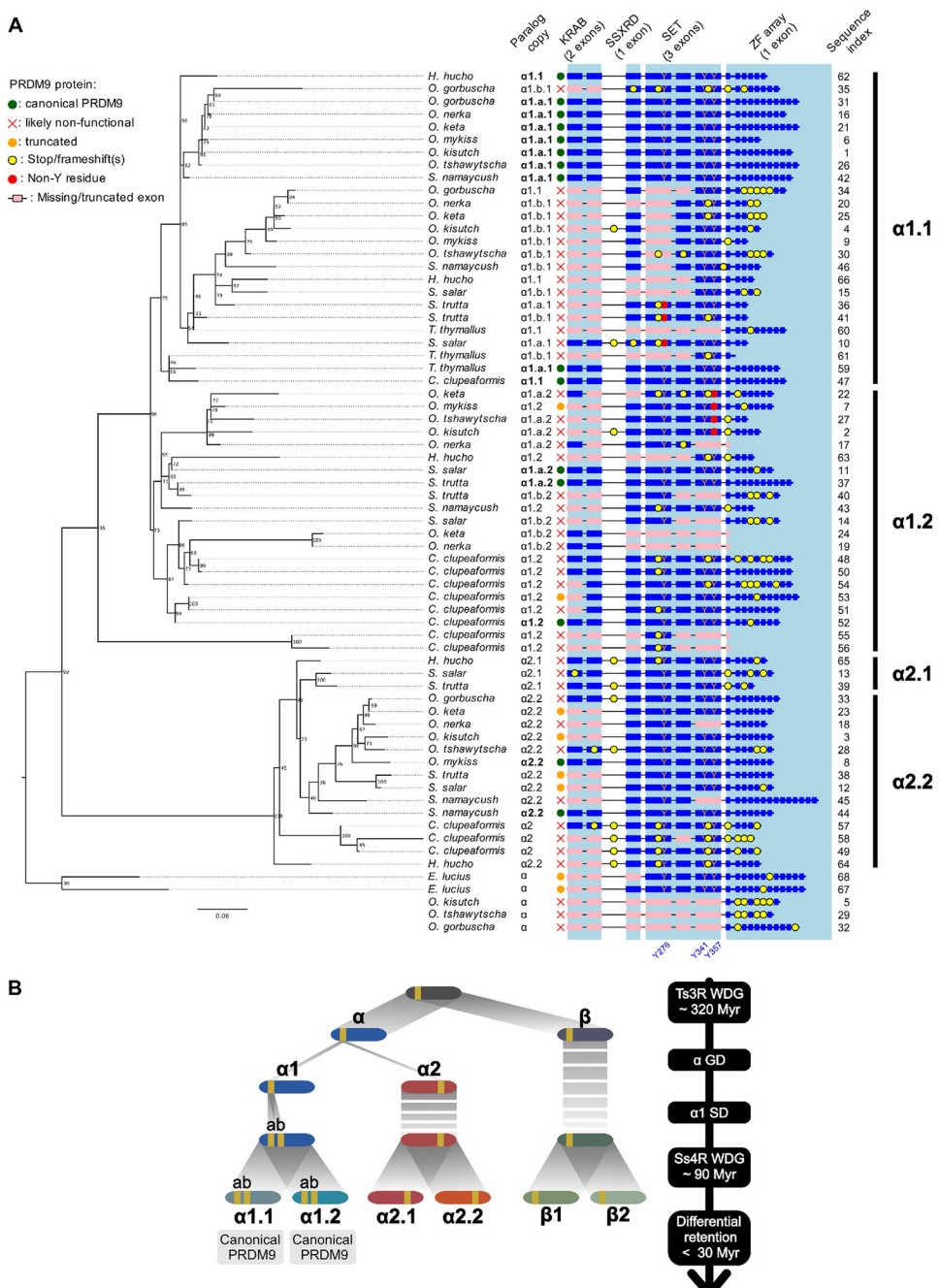

**Fig 1. *Prdm9* duplication history in salmonids.** (**A**) Phylogenetic tree of *Prdm9α* paralogs in 12 salmonids and northern pike (*Esox lucius*) as outgroup species. *Prdm9*β is shown in S1 Fig. The phylogenetic tree was computed on the concatenated 6 exons of the 3 canonical PRDM9 domains KRAB, SSXRD, and SET, with 1,000 bootstrap replicates (values shown). The columns, from left to right, indicate the (i) species name; (ii) annotated paralog copy (in bold: full-length copy without pseudogenization); (iii) *Prdm9* copy status. *Prdm9α* clusters into 2 main groups (*α1* and *α2*) that are divided in 2 subgroups (*α1.1/α1.2* and *α2.1/α2.2*). The scale bar is in unit of substitution per site. The right panel shows the coding potential of each paralog, and indicates the presence of frame-shifting mutations or stop codons, and of substitutions in the catalytic tyrosines of the SET domain (Y276, Y341, and Y357). Canonical (full length) Prdm9 proteins contain 4 key domains: KRAB (encoded by 2 exons), SSXRD (encoded by 1 exon), SET (encoded by 3 exons), and the ZF array (encoded by 1 exon). Complete exons are shown in blue. Missing or truncated exons are shown in pink. Other regions of the protein (upstream of the KRAB domain, and between KRAB and SSXRD) are encoded by additional exons (not shown here), that are not conserved between α1 and α2 clades. Paralogs were classified as "canonical PRDM9" if they contained all exons encoding the 4 key domains, without any frameshift/non-sense

mutation (at least up to the first ZF) [NB: some sequences contain frameshifts or non-sense mutations in the ZF array. This leads to a shortened ZF array, but does not necessarily impair the function of PRDM9]. Paralogs were classified as "likely non-functional" if they contained frameshifts or non-sense mutations, or if they missed at least 1 SET exon. Other cases were classified as "truncated." The 3 last α copies, belonging to *O. kisutch*, *O. tshawytscha*, and *O. gorbuscha*, have lost the 3 domains KRAB, SSXRD, and SET, but have kept their ZF exons, and were therefore added below the phylogenetic tree. The last column indicates the sequence indexes referring to the S1 Table with additional information on the corresponding copy. (**B**) Consensus history of *Prdm9* duplication events in salmonids. After the teleost-specific WGD (Ts3R WDG), the chromosomes of the common ancestor of teleosts were duplicated. Two ohnolog chromosomes arose from the one carrying the ancestral *Prdm9* locus: one carrying the *Prdm9α* copy and the other the *Prdm9β* copy. GD of the α paralog (referred to as *α1*) led to the appearance of a new α copy (*α2*) on another chromosome. The *α1* copy (becoming *α1.a*) then underwent an SD, generating a *α1.b* copy in tandem on the same chromosome. By this time, the β paralog had lost the KRAB and SSXRD domains. Lastly, the 4 copies were duplicated during the salmonids-specific Ss4R WGD, with the newly formed paralogs (annotated *α1.a.2*, *α1.b.2*, *α2.2*, *β2*) on ohnolog chromosomes. One full-length copy was retained in each species. The *Salmo* genus (*S. trutta* and *S. salar*) retained the *α1.2* copy, whereas all other salmonids retained the *α1.1* copy. A second full-length PRDM9 was also retained in *C. clupeaformis* (*α1.2*), *O. mykiss* (*α2.2*), and *S. namaycush* (*α2.2*). Ohnolog chromosomes are represented with similar color shades (i.e., blue, red, and green) and *Prdm9* locus in yellow. This global picture of the duplication events in the salmonid history does not show other independent lineage-specific duplication events and losses. The data and codes underlying this figure can be found in https://doi.org/10.5281/zenodo.11083953. GD, gene duplication; WGD, whole genome duplication; ZF, zinc finger.

the SET domain and without evidence of pseudogenization (Fig 1). In the *α1* clade, we detected on average 3.8 paralogs per genome, but each species retained only 1 full-length copy (corresponding to the *α1.a.1* paralog in *Thymallus*, *Oncorhynchus*, and *Salvelinus*, and to the *α1.a.2* paralog in the 2 *Salmo* species), except in *C. clupeaformis*, where both *α1.1* and *α1.2* are full length. Conversely, in the *α2* clade, we detected a full-length copy in only 2 species (*α2.2* in *O. mykiss* and in *Salvelinus namaycush*). Therefore, our results support the differential retention of functional *Prdm9α1* paralogs between salmonid lineages following the Ss4R WGD.

## High PRDM9 ZF array diversity in *O. mykiss* and *S. salar*

We analyzed the allelic diversity of the ZF array of the complete PRDM9α copy found in the Atlantic salmon *S. salar* (*α1.a.2*) and the rainbow trout *O. mykiss* (*α1.a.1*) (Fig 1). We identified 11 PRDM9 ZF alleles in 26 *S. salar* individuals and 7 alleles in 23 *O. mykiss* individuals (Fig 2A). The major allele had a frequency of 40% in *S. salar* and 35% in *O. mykiss*, and the 4 most frequent alleles had a cumulative frequency >80% in both species (Fig 2B). *S. salar* and *O. mykiss* alleles contained 5 to 10 and 7 to 15 ZFs, respectively. In both species, the last ZF of the arrays was probably not functional, because it lacked the conserved histidine involved in the interaction with a zinc ion required to stabilize the finger array (S3 Fig). As seen in other species [38], the 4 positions in contact with DNA (position −1, 2, 3, and 6 of the alpha helix) were highly variable among ZF units (Fig 2C). We characterized the proportion of total amino acid diversity at these DNA-binding residues among all different ZF units identified in each species following [19]. This proportion, which is sensitive to the rapid evolution at DNA-binding sites and to the homogenization at other amino acid positions due to concerted evolution between repeats within the array, was 0.49 in *S. salar* and 0.55 in *O. mykiss* (Fig 2C). These values were within the range reported for full-length PRDM9α in vertebrates [19]. The observed high level of allelic diversity and the pattern of amino acid diversity within the ZFs were consistent with the rapid and concerted evolution of the ZF array of the full-length *Prdm9* gene that characterizes PRDM9 copies involved in specifying meiotic recombination sites [19,54].

In addition to the full-length *α1* copy, we observed that the *α2.2* paralog is also strongly expressed in testes, both in *Oncorhynchus* and in *Salmo* genera. This paralog is full length in *O. mykiss* and *S. namaycush*, but in all other salmonids the KRAB domain of *α2.2* is missing, or pseudogenized (Figs 1 and S4). This phylogenetic pattern implies that *α2.2* lost its KRAB

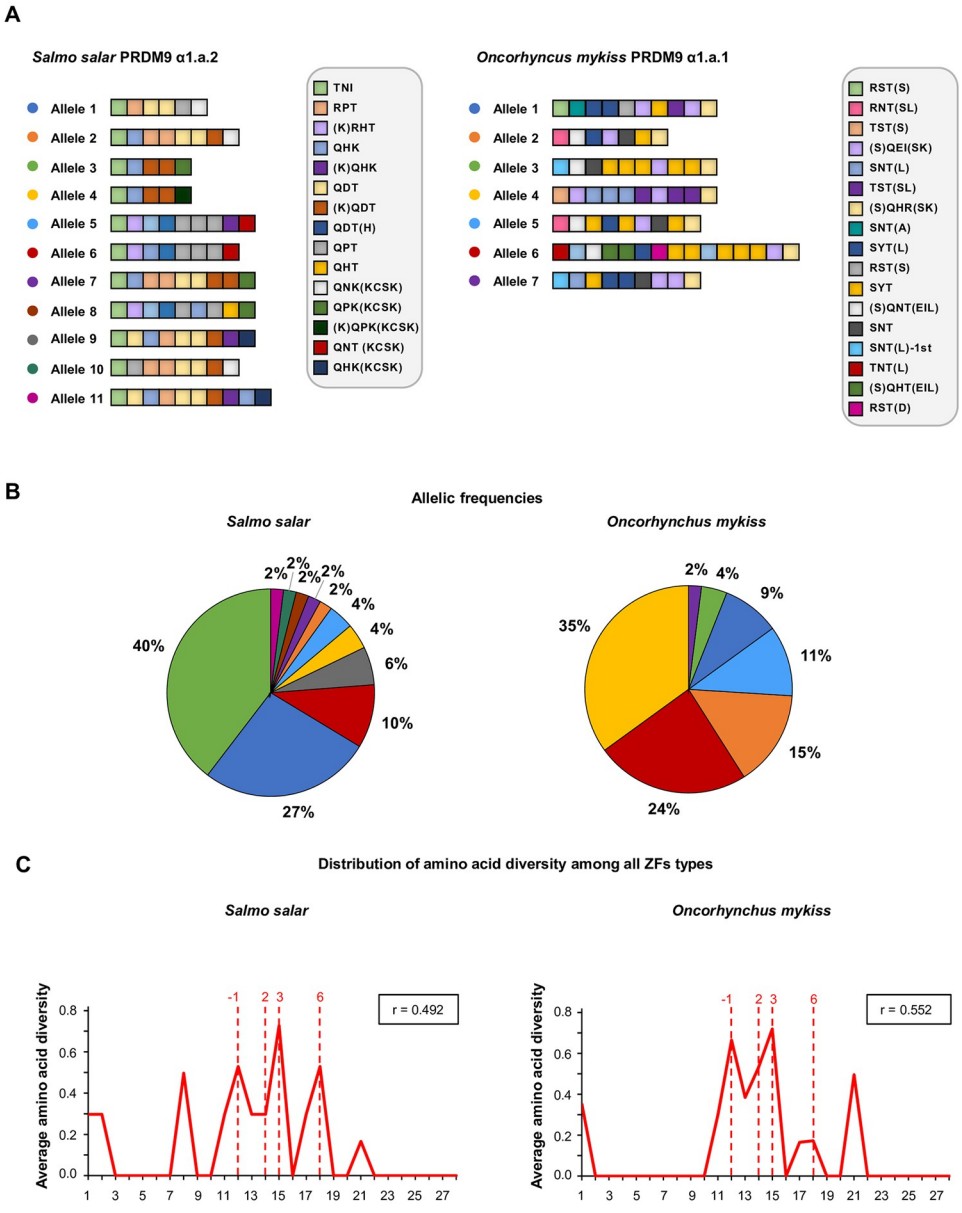

**Fig 2. Zinc finger allelic diversity of full-length PRDM9 in *S. salar* and *O. mykiss*.** (**A**) Structure of the identified PRDM9 alleles in *S. salar* PRDM9 α1.a.2 and *O. mykiss* PRDM9 α1.a.1. Colored boxes represent unique ZFs, characterized by the 3 amino acids in contact with DNA (3-letter code). Additional variations relative to the reference sequence are indicated in between brackets. The complete ZF amino acid sequences are shown in S3 Fig. (**B**) Frequencies of the alleles displayed in panel A among the 26 *S. salar* and 23 *O. mykiss* individuals in which *Prdm9* was genotyped. (**C**) Distribution of amino acid diversity among all unique ZFs found in the alleles shown in panel A, following a previously described methodology [19]. The amino acid diversity is plotted as a function of the amino acid position in the ZF array, from position 1 to position 28 (first and last residues) of a ZF. The ratio of amino acid diversity at the DNA-binding residues of the ZF array (−1, 2, 3, and 6), indicated as r, is shown in the upper box of each panel. The data underlying this figure can be found in https://doi.org/10.5281/zenodo.11083953 and in S7 Table. ZF, zinc finger.

domain several times independently, in different lineages. In *S. salar*, the allelic diversity of the ZF array in the truncated *Prdm9α2.2* was very low: in the 20 individuals analyzed, we observed 1 single allele where the array had 5 ZF units (S3 and S5A Figs). This is consistent with the hypothesis that KRAB-less PRDM9 homologs lost the capacity to trigger recombination

hotspots, and therefore, are no longer subject to the Red Queen dynamics [19]. The proportion of amino acid diversity at DNA-binding residues is relatively high among the 5 ZFs of this unique PRDM9 α2.2 allele (r = 0.471). The persistence of this signature of positive selection suggests that the functional shift associated with the loss of the KRAB domain is relatively recent. In *O. mykiss*, where PRDM9 *α2.2* is full length, we identified 5 *α2.2* alleles in 20 individuals, with 6 to 12 ZFs (S5A and S5B Fig). Some ZFs lost 1 amino acid, with unknown consequence on their DNA binding capacity (S3 Fig). In *O. mykiss*, the proportion of amino acid diversity at DNA-binding residues was lower in PRDM9 α2.2 (r = 0.367, S5C Fig) than in PRDM9 α1.a.1 (r = 0.552, Fig 2C). This observation, together with the relatively limited allelic diversity, suggests that *O. mykiss* PRDM9 *α2.2* is no longer subject to the Red Queen dynamics, and hence that it has lost its function of directing recombination, like the KRAB-less *α2.2* paralogs in other salmonids.

## PRDM9 specifies meiotic DSB hotspots in *O. mykiss*

To directly assess whether the full-length PRDM9α copy (hereafter PRDM9 unless otherwise specified) determines the localization of DSB hotspots, we investigated the genome-wide distribution of DMC1-bound ssDNA in *O. mykiss* testes by DMC1-SSDS (Fig 3A). DMC1 is a meiosis-specific recombinase that binds to ssDNA 3′ tails resulting from DSB resection. Therefore, meiotic DSB hotspots can be mapped by identifying fragment-enriched regions (i.e., peaks) in DMC1-SSDS data [30,33,69]. We detected several hundred peaks in the 3 rainbow trout individuals analyzed by DMC1-SSDS (616 peaks in TAC-1, 209 in TAC-3, and 1924 in RT-52). Differences in peak number may result from inter-sample differences in cell composition related to the testis developmental stage (see S1 Methods). In all 3 individuals, the DMC1-SSDS signal at DSB hotspots displayed a characteristic asymmetric pattern in which forward and reverse strand reads were shifted toward the left and the right of the hotspot center, respectively. This confirmed that the DMC1-SSDS peaks detected in rainbow trout were genuine meiotic DSB hotspots [30] (S6A Fig). The average width of DMC1-SSDS peaks was 1.5 to 2.5 kb, which is similar to what described in mice and humans [30,33]. The DSB hotspot density increased towards the chromosome ends, indicating that the U-shaped distribution of COs classically observed in male salmonids [70] is the result, at least in part, of a mechanism controlling DSB formation (S7A Fig).

Then, we tested whether the DSB hotspot formation was PRDM9-dependent by assessing the hotspot association with (i) specific *Prdm9* alleles; and (ii) sites enriched for both H3K4me3 and H3K36me3 due to PRDM9 methyltransferase activity [38,71]. The 3 individuals analyzed (only TAC-1 and TAC-3 for histone modifications) carried a functional *Prdm9* (i.e., *Prdm9α1.a.1*) with different genotypes. TAC-1 (*Prdm9*[1/5]) and TAC-3 (*Prdm9*[2/6]) did not share any *Prdm9* allele, whereas RT-52 (*Prdm9*[1/2]) shared 1 allele with each of them. In line with the hypothesis that PRDM9 specifies DSB hotspots, some DMC1-SSDS peaks were common to RT-52 and either TAC-1 or TAC-3 (see Fig 3A for examples). Specifically, the overlap between TAC-1 and RT-52 DSB hotspots (167 of the 616 TAC-1 hotspots, 27%), and between TAC-3 and RT-52 DSB hotspots (42 of the 209 TAC-1 hotspots, 20%) was substantial, whereas only 2 hotspots were shared by all 3 individuals (S6B Fig). The 55 DMC1-SSDS peaks shared by TAC-1 and TAC-3 may be artifactual because the forward and reverse strand enrichment distribution did not follow the typical asymmetric pattern of DSB hotspots, in contrast to the overlapping hotspots between TAC-1 and RT-52 and between TAC-3 and RT-52 (S7B Fig). The histone modifications H3K4me3 and H3K36me3 usually do not colocalize at the same loci because H3K4me3 is enriched at promoters and other genomic functional elements, whereas H3K36me3 is enriched within gene bodies. Indeed, at the peaks of H3K4me3 detected

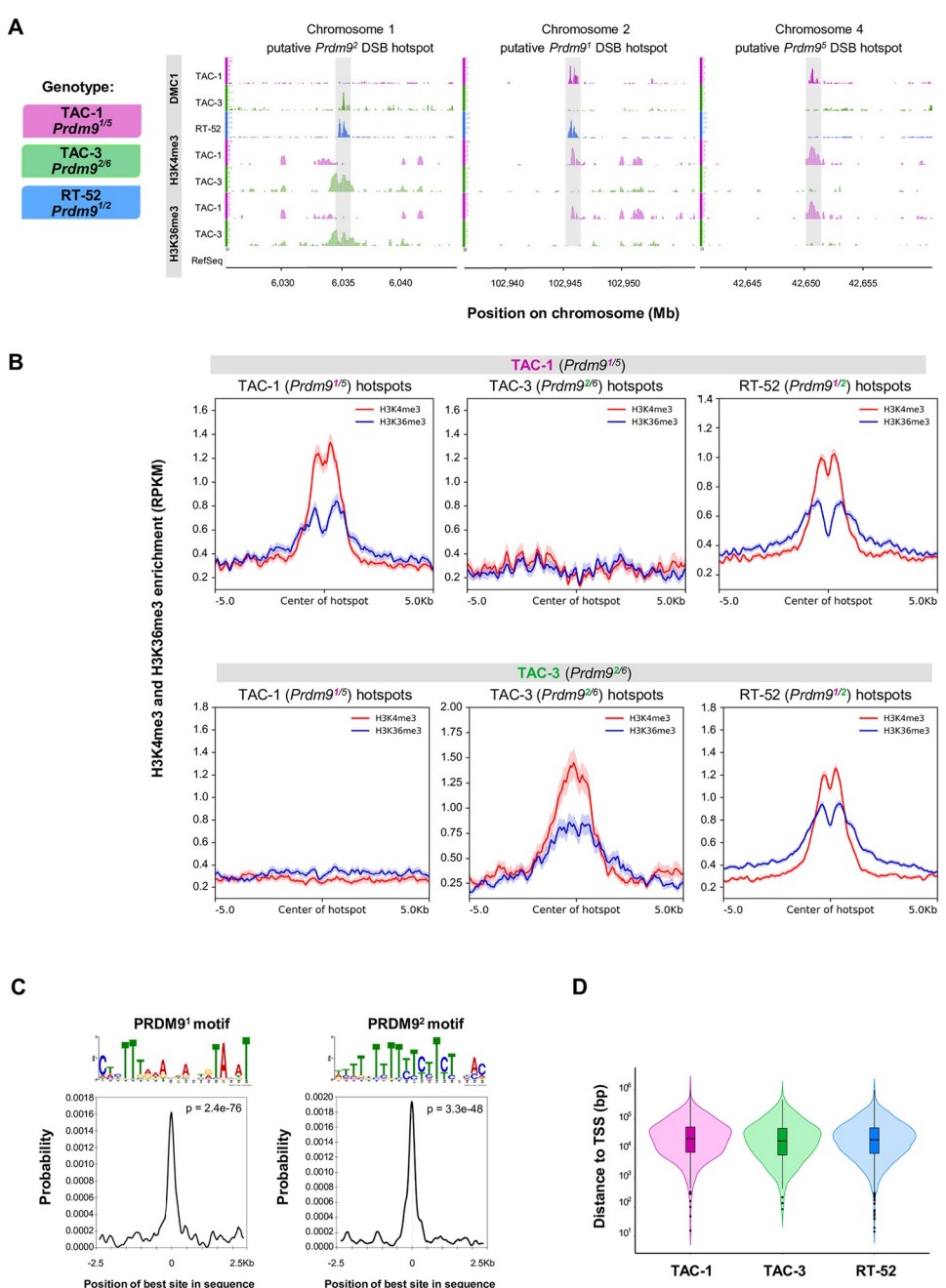

**Fig 3. Meiotic DSB hotspots are specified by full length PRDM9 in *O. mykiss*.** (**A**) DSB hotspots detected by DMC1-SSDS (DMC1), H3K4me3 and H3K36me3 in selected regions of the *O. mykiss* genome in testes from 2 or 3 (DMC1) individuals. (**B**) Average profile of H3K4me3 (red) and H3K36me3 (blue) ChIP-seq signal in TAC-1 (Prdm9$^{1/5}$) and TAC-3 (Prdm9$^{2/6}$) testes, at DSB hotspots detected in TAC-1 (Prdm9$^{1/5}$), TAC-3 (Prdm9$^{2/6}$), and RT-52 (Prdm9$^{1/2}$). (**C**) On top, the PRDM9 allele 1 (E = 5.1e-37) and allele 2 motifs (E = 1.2e-63) discovered in allele 1 (*n* = 300) and allele 2 DSB sites (*n* = 254) are shown. Below, the plots depict the distribution of hits for the PRDM9 allele 1 (left) and allele 2 (right) motifs at allele 1 and allele 2 DSB sites from the center of the sequence up to 2.5 kb of distance. The signal is smoothed by weighted moving average, and hits were calculated in a 250 bp sliding window. (**D**) Violin plot showing the distribution of DSB hotspots from TAC-1 (magenta), TAC-3 (green), and RT-52 (blue) relative to the TSS from RefSeq annotated genes. The data and codes underlying this figure can be found in https://doi.org/10.5281/zenodo.11083953 and https://zenodo.org/records/14198863. ChIP, chromatin immunoprecipitation; DSB, double-strand break; TSS, transcription start site.

in brain tissue where *Prdm9* is not expressed, no H3K36me3 enrichment was detected (S8A Fig). However, at the DSB hotspots mapped in TAC-1, an enrichment for H3K4me3 and H3K36me3 was detected in testis chromatin from TAC-1 but not from TAC-3 (Fig 3B, left panels) and reciprocally for the DSB hotspots mapped in TAC-3 (Fig 3B, central panels, S8B and S8C Fig). These observations are coherent with the PRDM9-dependent deposition of these histone modifications as TAC-1 and TAC-3 carry distinct *Prdm9* alleles. At the hotspots mapped in RT-52, an enrichment for H3K4me3 and H3K36me3 was detected in testis chromatin from TAC-1 or from TAC-3 (Fig 3B, right panels, S8B and S8C Fig) which is consistent with the presence of common *Prdm9* alleles between RT-52 and TAC-1 and between RT-52 and TAC-3. In addition, the RT-52 hotspots overlapping with TAC-1 are expected to be distinct from those overlapping with TAC-3, and specified by the *Prdm9¹* and *Prdm9²* alleles respectively. Indeed, the majority of RT-52 DSB hotspots were enriched for H3K4me3 either in testis chromatin from TAC-1 or in TAC-3, but not in both (S9B Fig). A similar effect for H3K36me3 could not be concluded due to the high level of PRDM9-independent H3K36me3 at a fraction of the sites (S9A and S9B Fig).

## Population genomic landscapes of recombination

The DMC1-SSDS approach allows analyzing DSB distribution in a given male individual, but is thus restricted to one sex and does not provide information on the outcome of recombination events (CO or NCO). To get a more general picture of the genome-wide recombination landscapes and their evolution, we computed LD-based genetic maps in 3 salmonid species: coho salmon (*O. kisutch*), rainbow trout (*O. mykiss*), and Atlantic salmon (*S. salar*). In *S. salar*, we analyzed 3 populations: North Sea (NS), Barents Sea (BS), and Gaspesie Peninsula (GP). For comparison, we also reconstructed the LD-based recombination map of European sea bass (*Dicentrarchus labrax*) that carries the KRAB-less *Prdm9β* gene, but lacks a full-length *Prdm9α*.

The population-scaled recombination landscapes showed consistent broad-scale characteristics between *O. kisutch*, *O. mykiss*, and the 3 *S. salar* populations. The genome-wide population recombination rate ranged from 0.0032 (in units of $\rho = 4N_e r$ per bp) in *O. kisutch* to 0.012 in *O. mykiss*, with intermediate values in *S. salar* populations (Table 1). At the intra-chromosomal level, 100 kb smoothed recombination landscapes showed a general increase towards the chromosome ends, up to a 6-fold increase in *S. salar* (S10 Fig). This U-shape pattern mirrored the chromosomal distribution of DSB hotspots in male rainbow trout (S7A Fig).

**Table 1. Summary of fine-scale recombination rate variations in 2 kb windows, hotspot detection, effective population size ($N_e$), recombination rate obtained from pedigree-based sex-averaged genetic maps [67,75–77], and recombination to mutation rate ratio for populations of *O. kisutch*, *O. mykiss*, *S. salar* (only the NS population is shown), and *D. labrax*.** $N_e$ ranges were estimated based on the mean nucleotide diversity measured in population resequencing datasets and mutation rates reported in fish and human (see Methods for details). $\mu/r$ ratio ranges were calculated using $r$ obtained from pedigree-based genetic maps.

| | *O. kisutch* | *O. mykiss* | *S. salar* | *D. labrax* |
|---|---|---|---|---|
| Genome wide recombination rate ($\rho$/bp) | 0.0032 | 0.012 | 0.0085 | 0.039 |
| Cumulative amount of recombination in the 20% most recombining regions | 90.1% | 89.1% | 98.1% | 84.6% |
| Number of hotspots | 22,948 | 21,145 | 17,064 | 7,897 |
| Fraction of recombination in hotspots | 36.7% | 19.3% | 18.3% | 26.5% |
| Fraction of the genome occupied by hotspots | 2.7% | 2.1% | 1.3% | 1.9% |
| Hotspot density (per Mb) | 13.6 | 10.8 | 6.8 | 9.6 |
| $N_e$ | [28,220–141,99] | [80,302–401,512] | [18,51–90,254] | [22,600–113,000] |
| $r$ (in cM/Mb) | 2.24 | 1.31 | 1.99 | 2.77 |
| $\mu/r$ | [0.09–0.45] | [0.15–0.76] | [0.1–0.5] | [0.07–0.36] |

The fine-scale analysis of the genomic landscapes also showed highly heterogeneous recombination rates within 2 kb windows (Table 1 and S11 Fig). In each population, the local variation in recombination rate was of several orders of magnitude (S2 Table). On average, 90% of the total recombination appeared to be concentrated in 20% of the genome, a higher rate than what was observed in human and chimpanzee [29,72] and slightly higher than what we observed in sea bass (Tables 1 and S2 and S12 Fig). This heterogeneity was largely driven by the presence of recombination hotspots. Based on the raw LD maps reconstructed at each SNP interval, we confirmed that the size of most (>80% on average) salmonid hotspots was <2 kb (S13 Fig and S2 Table). Therefore, we performed the rest of our analysis using the hotspots called within 2 kb windows. The total number of called hotspots per species ranged from 17,064 in *S. salar* to 22,948 in *O. kisutch*, with hotspot density values similar to those in sea bass and also humans, mice, and snakes [60,72,73]. The proportion of total recombination cumulated in hotspots ranged from 17% in *S. salar* to 36% in *O. kisutch*, while occupying less than 3% of the genome (Table 1).

Then, we compared the LD-based recombination landscape of *O. mykiss* and the location of DSB hotspots mapped by DMC1-SSDS (pooling peaks from the 3 samples). We found that 6.7% of DMC1-SSDS peaks overlapped with the LD-based hotspots, which is more than expected by chance (S14A and S14B Fig). This weak overlap was comparable with that observed in *Mus musculus castaneus* where 12% of DSB hotspots overlap with LD-based hotspots [74]. We also found that in these shared peaks, population recombination rates were significantly higher than in non-shared LD-based or DSB hotspots and the rest of the background landscape (Kruskal–Wallis test $p$-value <0.05, Wilcoxon post hoc test < 0.05, S14C Fig).

## Recombination hotspots are located away from TSSs

In species that lack full-length PRDM9, recombination hotspots are expected to be located in open-chromatin regions, such as unmethylated CGI-associated promoters and/or constitutive H3K4me3 sites [18,19,22,25,27], unlike in species like mice, where PRDM9 targets regions away from these genetic elements [30]. To test whether PRDM9$\alpha$ plays a similar role in salmonids, we first examined how DSB hotspots were distributed relative to TSSs in rainbow trout. We found that the percentage of DSB hotspots overlapping with TSSs was either not different or lower than expected by chance (4.5% and 5.3% versus 7.6% for TSSs of coding and non-coding genes; S3 Table and S6C Fig). Moreover, the vast majority of DSB hotspots mapped several kb or more away from the closest TSS (Fig 3D). Therefore, DSB hotspots, at least those strong enough to be detected by our DMC1-SSDS assay, did not localize at TSSs.

We then examined how population recombination rates were distributed relative to TSSs that overlapped or not with CGIs, by comparing the 3 salmonid species to sea bass that only has a truncated PRDM9$\beta$ protein. Although the criteria classically used to predict CGIs in mammals and birds are not appropriate for teleost fish where CGIs are CpG-rich but have a low GC-content [78,79], we could predict TSS-associated CGIs in fish genomes simply based on their CpG content (see S1 Analysis). Sea bass (truncated PRDM9$\beta$) showed a high level of recombination at promoter regions, with a strong 3-fold enrichment of recombination at TSSs associated with CGIs (Fig 4A), as reported in birds [25]. Conversely, in salmonid species (full-length PRDM9), recombination rate varied little between TSSs and their flanking regions (at most 1.2-fold enrichment). Specifically, at CGI-associated TSSs, recombination rate tended to be lower than at other TSSs (Fig 4A and 4B). Moreover, hotspots overlapping with TSS represented <5% of all hotspots in the 3 salmonid populations and up to 21% in sea bass (Fig 4C). The analysis of other genomic features showed little variation in recombination rate and hotspot density, with similar levels in genes, introns, exons, TEs, and CGIs compared with

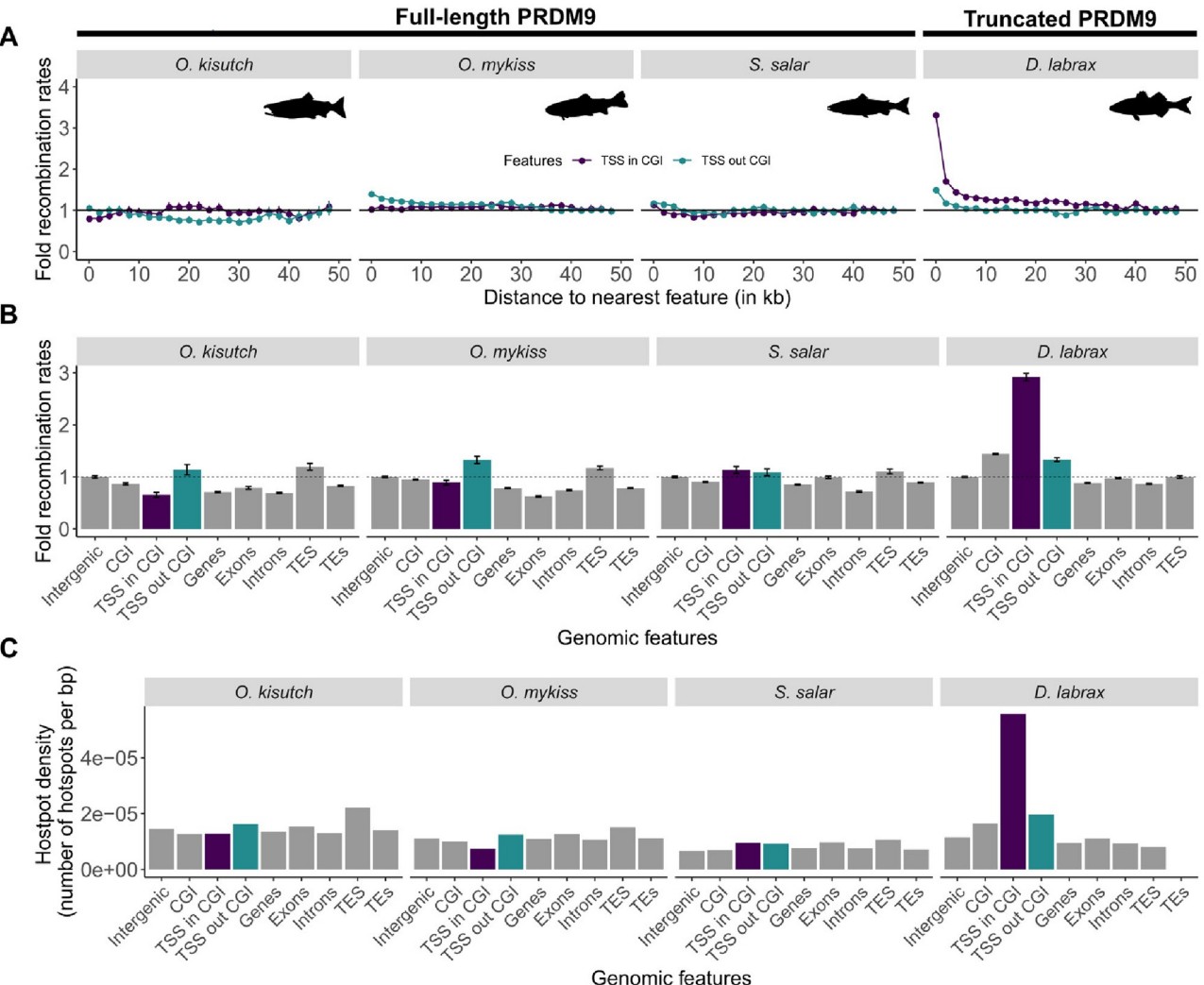

**Fig 4. Recombination rates at genomic features.** The recombination rates at different genomic features are shown for *O. kisutch*, *O. mykiss*, and *S. salar* (NS population), and compared to those of sea bass (*D. labrax*) that lacks a full-length PRDM9 copy. (**A**) Fold recombination rates (scaled to the average recombination rate at 50 kb from the nearest feature) according to the distance to the nearest TSS (overlapping or not with a CGI). (**B**) Fold recombination rates (scaled to the average recombination rates in intergenic regions) at the indicated genomic features. (**C**) Hotspot density at the indicated genomic features. TSS in and out CGI are shown in purple and blue, respectively. The data and codes underlying this figure can be found in https://doi.org/10.5281/zenodo.11083953. NS, North Sea; TSS, transcription start site.

intergenic regions (Fig 4B and 4C). We observed only a very small increase in recombination rate at TSSs that did not overlap with CGIs and TESs in *O. kisutch* and *O. mykiss*. Therefore, our results indicated that salmonid recombination events do not concentrate at promoter-like features overlapping with CGIs, as already shown in primates and in the mouse.

We also examined other genomic correlates and features that might influence population recombination rate variation at different levels of resolution. As expected from the joint effect of the local effective population size ($N_e$) on both nucleotide diversity and population recombination rate, SNP density was positively correlated with the ρ averaged at the 100 kb scale, although this trend was not significant in *O. mykiss* (S15A Fig). More locally, we also observed an increase in SNP density in the 10 kb surrounding recombination hotspots (S16A Fig). These positive relationships could be amplified by a direct mutagenic effect of recombination

during DSB repair, and a more pronounced erosion of neutral diversity in low-recombining regions due to linked selection [29,33,80–82]. However, we cannot exclude the possibility that the accuracy of the recombination rate estimate depends on SNP density [83], leading to possible confounding effects.

In mammals, GC-biased gene conversion causes an increase in GC-content at recombination hotspots [33,40,84,85]. Conversely, in all the 5 salmonid populations analyzed, except the GP population, GC content tended to decrease close to hotspots (S16B Fig). At a larger scale (i.e., 100 kb), we observed significant positive correlations between GC content and recombination rates (S15B Fig). However, these correlations were very weak, suggesting that GC-biased gene conversion has a very small impact in salmonids compared with mammals and birds [85].

The salmonid genomes contain a high density of TEs (covering approximately 50% of the genome), among which Tc1-mariner is the most abundant superfamily (>10% of TEs) [66]. It is not known whether Tc1-mariner transposons influence the estimation of recombination rates. Our TE analysis identified between 47.37% and 52.26% of interspersed repeats in *O. kisutch* and *S. salar*, respectively, and showed that 12.48% to 14.7% of the genome was occupied by Tc1-mariner elements (S4 Table). TEs and intergenic regions showed similar average recombination rates and hotspot density (Fig 4B and 4C). Recombination rates tended to slightly increase with TE density at the larger scale, except in *O. mykiss* for which we observed the opposite relation (S15C Fig), without any strong effect of the TE superfamilies (S17 Fig). As recombination rates and hotspot density at TEs were globally comparable to those at intergenic regions (Fig 4B and 4C), TEs and among them Tc1-mariner elements did not seem to be characterized by extreme recombination values that may have affected our recombination rate estimations.

Lastly, residual tetrasomy resulting from the salmonid WGD event at approximately 90 Mya (Ss4R) [64,66] is observed at several chromosome regions characterized by increased genomic similarity between ohnologs. This could also affect the inference of LD-based recombination rates. Such regions have been identified in *O. kisutch*, *O. mykiss*, and *S. salar* [66–68]. We tried to filter non-diploid allelic variation from chromosomes showing residual tetrasomy, and we also controlled their effect by comparing their recombination patterns with those of fully re-diploidized chromosomes. Overall, we found <2-fold increase of the mean recombination rate in chromosomes containing tetraploid regions (S18A Fig). This was mostly explained by the local increase towards the end of chromosomes with residual tetraploidy compared with fully re-diploidized chromosomes, an effect that was especially pronounced in *O. mykiss* (S18B Fig). Nevertheless, recombination rates behaved similarly in function of the distance to the nearest promoter-like feature in the 2 chromosome sets, and rate variations were similar between genomic features (S19 Fig). Overall, chromosomes containing regions with residual tetraploidy and re-diploidized chromosomes showed similar recombination patterns.

## Rapid evolution of recombination landscapes

Another key feature of the mammalian system is the rapid evolution of PRDM9-directed recombination landscapes due to self-induced erosion of its binding DNA motif and rapid PRDM9 ZF evolution [29,32,34]. To determine whether this feature was present also in salmonids, we compared the location of recombination hotspots in the 2 *Oncorhynchus* species and in 2 geographical lineages and 2 closely related populations of *S. salar*. We estimated that only 6.2% of hotspots (*n* = 1,298) were shared by *O. kisutch* and *O. mykiss*, which diverged from their common ancestor about 16 Myr ago [86]. Although this value was significantly higher than expected by chance (S20A Fig), there was almost no increase in recombination rate at the

orthologous positions of hotspots in the 2 species (Fig 5A). Similarly, the 2 genetically differentiated lineages of *S. salar* only shared 10.3% (GP versus BS, $F_{ST} = 0.26$, $n = 1,793$) and 11.2% (GP versus NS, $F_{ST} = 0.28$, $n = 1,671$) of their hotspots, with a weak recombination rate increase at the alternate lineage hotspots (Figs 5B, S20B and S20C). Conversely, the 2 closely related BS and NS *S. salar* populations ($F_{ST} = 0.02$) shared 26.3% of their hotspots ($n = 4,421$), which was much more than expected by chance (Figs 5C and S20D). In addition, recombination rate at NS hotspots in the BS population showed a 5-fold increase (and reciprocally), reflecting high correlation between BS and NS recombination landscapes (Spearman's rank coefficient $>0.7$, $p$-value $<0.05$; S21 Fig). Overall, these analyses revealed a rapid evolution of hotspot localization between species and also between geographical lineages of the same species. Only closely related populations shared a substantial fraction of their hotspots. This overlap probably reflects their similar genetic background (low $F_{ST}$), and in particular, the fact that they may share similar sets of *Prdm9* alleles recognizing common binding DNA motifs.

## Motifs enriched at hotspots show signs of erosion

A landmark of the PRDM9-dependent hotspots identified in mammals is the presence of DNA motifs, as a consequence of the sequence-specificity of the PRDM9 ZF domain [32]. Therefore, we investigated the presence of PRDM9 allele-specific DNA motifs enriched at hotspots in salmonids. We first searched for potential PRDM9 binding motifs in rainbow trout, focusing on RT-52 ($Prdm9^{1/2}$) DSB-based hotspots. As the $Prdm9^1$ allele is present in RT-52 and TAC-1 ($Prdm9^{1/5}$), we defined a subset of RT-52 DSB hotspots presumably specified by PRDM9$^1$, based on their overlap with H3K4me3/H3K36me3 peaks in TAC-1 ($n = 300$). Similarly, we defined a subset of DSB hotspots enriched in putative targets of the PRDM9$^2$ allele, which is present also in TAC-3 ($Prdm9^{2/6}$) ($n = 254$). We identified 2 consensus motifs: one strongly enriched in PRDM9$^1$ DSB hotspots and the other in PRDM9$^2$ DSB hotspots (Fig 3C). Consistent with the *Prdm9* genotypes of the 3 rainbow trout samples, both motifs were enriched at RT-52 DSB hotspots. The PRDM9$^1$ motif was also enriched in TAC-1 DSB hotspots and the PRDM9$^2$ motif in TAC-3 DSB hotspots (S22 Fig). Moreover, the PRDM9$^1$ motif was co-centered with DSB hotspots only in RT-52 (Fisher's test, $p = 8.5 \times 10^{-196}$) and TAC-1 ($p = 7.7 \times 10^{-27}$), while the PRDM9$^2$ motif was co-centered with DSB hotspots in RT-52 ($p = 3.3 \times 10^{-97}$) and TAC-3 ($p = 1.7 \times 10^{-5}$) (S23 Fig). These 2 consensus motifs were also significantly enriched at LD-based hotspots (S22 Fig). Particularly, the motif targeted by PRDM9$^1$ was enriched at the center of LD-based hotspots (S23 Fig), suggesting that this allele (or closely related alleles that recognize similar DNA sequences) has been quite frequent during the recent history of the wild population under study.

As PRDM9-binding DNA motifs are allele specific, the sharing of *Prdm9* alleles between populations should lead to shared motif enrichment at shared LD-based hotspots. Therefore, we looked for enrichment of potential 10 to 20 bp motifs in the population-specific and shared hotspots of the 3 *S. salar* populations. Of note, as LD-based hotspots reflect the population-scaled recombination rate, they may result from the activity of multiple PRDM9 variants that can hinder the discovery of targeted motifs. Nevertheless, after filtering candidate motifs (S24A Fig), we found a motif that was enriched in 12% of the hotspots of the NS population and 8.9% of the BS population, and in 15.6% of their shared hotspots (Fig 5D). Overall, the recombination rates at hotspots overlapping with this 12 bp motif were significantly higher than those at other hotspots (Student's *t* test $p$-value $<0.05$; Figs 5E, S24B and S24C). This suggests that the detected motif is targeted by a frequent PRDM9 variant shared by the 2 closely related NS and BS populations, possibly originating from their common ancestral variation.

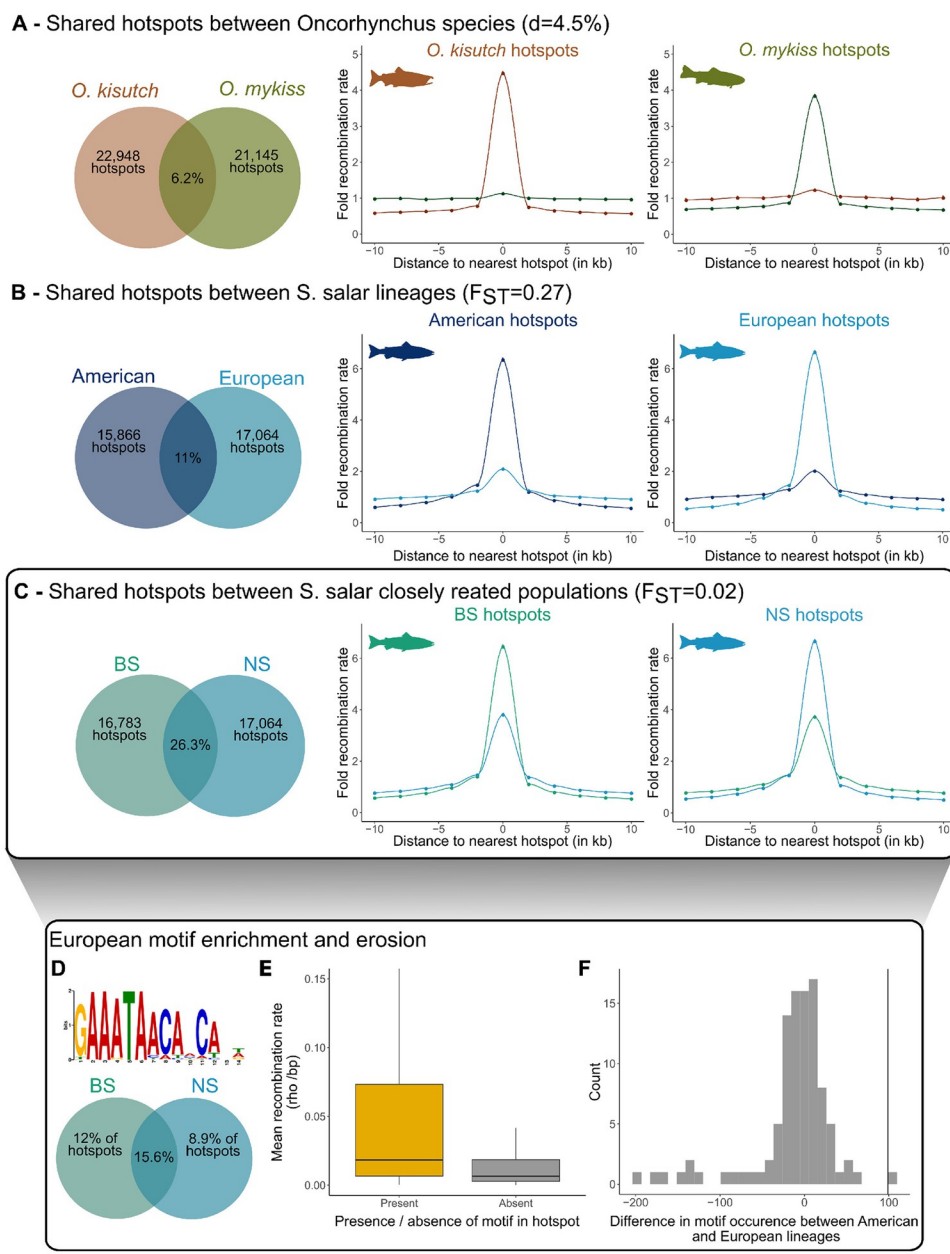

**Fig 5. Recombination hotspots shared between populations and motif enrichment.** In panels (A–C), the Venn diagrams (left) show the percentages of recombination hotspots shared between pairs of taxa, and the graphs (middle and right) show the recombination rates around hotspots and at orthologous loci in the 2 taxa, for the 2 *Oncorhynchus* species (**A**), the American (GP population) and European (BS and NS populations) *S. salar* lineages (**B**), and between the 2 closely related European *S. salar* populations (BS and NS) (**C**). The percentage of shared hotspots was calculated using the number of hotspots in the population with fewer hotspots as the denominator. (**D**) Motif found enriched in the hotspots identified in the European populations of *S. salar* (BS and NS). The Venn diagram shows the percentages of population-specific and shared hotspots where the motif was found. (**E**) Mean recombination rate at shared hotspots (between the BS and NS populations) that harbor ($n$ = 936 hotspots) or not ($n$ = 3,485 hotspots) the detected motif. The recombination rate was significantly higher at hotspots with the motif (Student's $t$ test $p$-value <0.05). (**F**) Motif erosion in the European *S. salar* populations. The vertical line represents the observed difference in the occurrence of the motif in panel D between the American and European lineages. The null distribution (in gray) shows the difference for 100 random permutations of the motif. The data and codes underlying this figure can be found in https://doi.org/10.5281/zenodo.11083953. BS, Barents Sea; GP, Gaspesie Peninsula; NS, North Sea.

PRDM9-associated hotspot motifs undergo erosion in mammals due to biased gene conversion [32,39,40]. Therefore, we tested whether the identified 12 bp motif showed signs of erosion in European *S. salar* populations. By comparing the number of motifs present in the available long-read genome assemblies from 7 European and 5 North American Atlantic salmon genomes (accession numbers in S1 Methods), we found a 2.97% reduction in the mean number of motifs in the European genomes (mean Europe = 3,230 versus mean North America = 3,329). This level of erosion was significant and not explained by differences in assembly sizes, as revealed by count comparisons on collinear blocks, obtained following 100 random permutations of the motif (Fig 5F). Therefore, the enriched motif shared by the NS and BS populations was partially eroded in the European lineage, as predicted by the Red Queen model of PRDM9 evolution.

## Discussion

To determine whether the PRDM9 functions characterized in humans and mice are shared by other animal clades or whether they correspond to derived traits, we investigated the evolution and function of full-length *Prdm9* in salmonids using phylogenetic, molecular, and population genomic approaches. These analyses allowed us to determine the evolutionary history of *Prdm9* GD and loss, the diversity of the PRDM9 ZF array, the historical sex-averaged recombination map in several populations, the locations of meiotic DSB sites in spermatocytes, their chromatin environment, and the presence of conserved motifs and their erosion. Collectively, these analyses led us to conclude that PRDM9 triggers recombination hotspot activity in salmonids through a mechanism similar to that described in mammals.

### PRDM9 specifies recombination sites in salmonids

Our conclusion is based on several pieces of evidence. First, we showed in *O. mykiss* that DSB hotspots, detected by DMC1-SSDS, are enriched for both H3K4me3 and H3K36me3. We provide evidence that hotspot localization is determined by PRDM9 ZFs because the location of DSB hotspots and the associated H3K4me3 and H3K36me3 modifications varied in function of the *Prdm9* ZF alleles present in the tested individuals (Fig 3A and 3B). Consistent with this interpretation, we identified DNA motifs enriched at DSB sites. Thus, in salmonids, PRDM9 retained its DNA binding and methyltransferase activities and the capacity to attract the recombination machinery at its binding sites. Comparison of DSB hotspots detected by DMC1-SSDS with LD-based CO hotspots in *O. mykiss* showed a limited, but significant overlap (S14 Fig). One should note that the quantitative level of DMC1 enrichment assayed by DMC1-SSDS can be influenced by the efficiency of DSB repair. If hotspots have variable efficiencies of repair, the quantitative correlation between DSB and LD hotspots could therefore be reduced. We also identified a DNA motif enriched at DSB hotspots targeted by *Prdm9*[1] that was also enriched at the center of strong CO hotspots detected in the LD-based recombination map (S23 Fig). The overlap between hotspots is compatible with the presence of a common *Prdm9* allele(s) between the individuals tested and the prevalent *Prdm9* allele(s) during the history of the populations analyzed. However, as the population-scaled recombination landscape in *O. mykiss* has been shaped by a diversity of alleles, not necessarily represented in the 3 studied individuals, the overall hotspot overlap was low. Similar variations in the recombination landscapes driven by multiple PRDM9 alleles have been described in mouse, chimpanzee, and human populations [29,35,36,74,87–89]. In the mouse, PRDM9 can suppress the recombination activity at chromatin accessible regions [30]. Here, we observe that in salmonids the presence of PRDM9-dependent hotspots is correlated with a lack of elevated recombination rate at regulatory regions (CGI, TSS, or TES) (Figs 3D and 4, and S6). We suggest that this may reflect

an active suppression or competition between the 2 types of hotspots, similarly to what has been observed in mice [30], since TSS and CGIs have elevated recombination rates in *D. labrax* (Fig 4).

## Comparison of recombination landscapes between vertebrates with or without a full-length PRDM9

In the absence of PRDM9, hotspots occur in accessible regions of the chromatin such as promoters, enhancers, or other regulatory regions and CGIs [18,19,22,25,30,57,58]. In addition to the change in distribution, differences in hotspot number have been detected between PRDM9-dependent and -independent contexts. By DMC1-SSDS, a greater number of hotspots was detected in *Prdm9KO* mice or rats [30,58]. However, this should be interpreted with caution as hotspot detection also depends on the half-life of DMC1 at DSB sites. A longer half-life of DMC1 in *Prdm9KO* may also account for an increase of detected hotspots. By LD-based approach, the number of hotspots detected is 2 to 3 times higher in the 3 salmonid species than in *D. labrax* (Table 1), but this difference is mainly explained by their larger genome sizes (1.7 to 2.5 Gb, compared to 0.6 Gb for *D. labrax*). To get a broader view of the impact of PRDM9 on vertebrate fine-scale recombination landscapes, we combined our data with previously published LD-based maps, thus resulting in a dataset of 18 species (4 birds, 7 mammals, 1 snake, and 6 teleost fish; 10 species with a full-length PRDM9, and 8 without; S5 Table). On average, the hotspot density is about 2 times higher in genomes with a full-length PRDM9 (8.6 hotspots/Mb) than without (4.4 hotspots/Mb; *t* test *p*-value = 0.056). However, it is difficult to directly compare these numbers, because different studies used different criteria to define hotspots. To get a more comparable estimator of the heterogeneity of recombination landscapes, we measured the fraction of recombination events occurring in the 20% of the genome with the highest recombination rate. On average, in genomes with a full-length PRDM9, 84% of recombination is concentrated in 20% of the genome, compared to 70% in genomes without (*t* test *p*-value = 0.011). Data from more species would be necessary to control for phylogenetic inertia. However, this preliminary observation suggests that recombination is more concentrated into hotspots in species having a full-length PRDM9. Of note, the LD-based approach measures the population-scaled recombination rate, integrated over many generations, and hence is expected to reflect the historical diversity of PRDM9 alleles. It is therefore likely that in species with PRDM9, the recombination landscapes of individuals are even more heterogeneous than what can be measured by the LD-based approach.

In addition to the localization of recombination shaped by PRDM9, we detected a higher recombination activity at telomere-proximal regions when measuring DSB activity and LD, consistent with the recombination activity measured in *S. salar* pedigree-based linkage maps [70]. We infer that this effect is PRDM9-independent because the putative PRDM9 motifs in *O. mykiss* (derived from DMC1-SSDS) and *S. salar* (derived from LD-based hotspots) did not show such biased distribution (S25 Fig). Of note, the increase in recombination rate towards telomeres is more pronounced in the 3 salmonids (3- to 6-fold) than in *D. labrax* (about 2-fold), but it is of the same order as in another teleost fish, the three-spined stickleback [61], which only has a truncated KRAB-less PRDM9ß (S10 Fig). We hypothesize that in salmonids, some additional factor(s) might modulate PRDM9 binding or any other step required for DSB activity along chromosomes. This telomere-proximal effect appears to be a conserved property, but of variable strength between sexes and among species, independently of the presence/absence of *Prdm9* [25,90,91].

## PRDM9 evolutionary instability

Similarly to the pattern reported in mammals [32,92,93], we found an outstanding diversity of PRDM9 ZF alleles in *O. mykiss* and *S. salar* and signatures of positive selection for ZF residues that interact with DNA, specifically in the full-length PRDM9 paralog (α1.a.1 and α1.a.2, respectively) (Fig 2). This suggests that full-length PRDM9 in salmonids could be involved in a Red Queen-like process, as documented in mammals, whereby the ZF sequence responds to a selective pressure arising from the erosion of PRDM9 binding motifs [39,40,47–50]. Consistent with this hypothesis, we found almost no overlap of LD hotspots in the 2 *Oncorhynchus* species we compared. A similar comparison performed in 3 *S. salar* populations revealed that the percentage of shared hotspots decreased with the increasing genetic divergence (Fig 5). The 26.3% overlap in hotspot activity we detected in the 2 Norwegian populations could reflect the existence of shared *Prdm9* alleles. On the other hand, the European and Northern American salmon populations, which belong to 2 divergent lineages, may not share the same *Prdm9* alleles and as a possible consequence, only have 10.5% of common hotspots. Such patterns of population-specific hotspots and partial overlaps have been observed also in mouse populations [35], great apes [94], and humans [95]. However, hotspot overlapping is always well below the 73% of shared hotspots between zebra finch and long-tailed finch that do not carry *Prdm9* [25]. Further support for a *Prdm9* intra-genomic Red Queen process in Atlantic salmon came from the detection of an enriched motif in 20% of the hotspots shared by the NS and BS populations. As this motif is likely to be the target of an active *Prdm9* allele in European populations, the average 3% decrease in total copy number in European populations compared with North American populations is indicative of ongoing motif erosion.

## Functional divergence of PRDM9 paralogs

Another intriguing pattern revealed by our study is the complex duplication history of the *Prdm9* gene in salmonids, shaped by WGD events and by gene and/or SDs. Some of these duplications led to functional innovations. Notably, the 2 major PRDM9 clades (α and ß) resulted from the Ts3R WGD in the ancestor of teleost fish [19]. PRDM9-ß lacks the KRAB and SSXRD domains, and is mutated at the catalytic residues of its SET domain [19]. The function of PRDM9ß has not been characterized, but the fact that this protein is strongly conserved across teleost fish, including salmonids that have a full-length PRDM9α (S1 Fig), implies (i) that it is functional; and (ii) that its function is not redundant with that of the canonical full-length PRDM9. Interestingly, the salmonid-specific WGD generated 2 PRDM9ß paralogs (ß1 and ß2) that are well conserved across all salmonids (S1 Fig), which indicates that they both are under purifying selection.

In contrast to the conservation of PRDM9ß paralogs, the duplications of PRDM9α genes led to many copies that are truncated or show evidence of pseudogenization (Fig 1). The first event of GD generated the α1 and α2 clades. All salmonids (12/12) have one full-length copy in the PRDM9α1 clade (in the subclade *α1.1* in some species, *α1.2* in others, except *C. clupeaformis* that has retained both *α1.1* and *α1.2*). Conversely, only 2 species have retained a full-length PRDM9α2 paralog (*α2.2* in *O. mykiss* and *S. namaycush*). In all other species, the KRAB domain of *α2.2* is missing or pseudogenized (Fig 1). The analysis of published RNAseq data sets showed that PRDM9*α2.2* is expressed at high level in testis, both in *O. mykiss* (where it is full length) and in *O. kisutch* and *S. salar* (where it is truncated). The SET domain of *α2.2* contains the 3 conserved tyrosine residues important for methyltransferase catalytic activity [96] (Figs 1 and S4). But PRDM9*α2.2* genes show little diversity at their ZF domain, which suggests that unlike full-length PRDM9α1, it is probably not involved in directing recombination. It is possible that those paralogs contribute together with the full-length PRDM9 to hotspot

activity. For example, they may have retained putative protein interaction properties through the SSXRD domain or some zinc fingers, and may also be able to oligomerize with PRDM9 as proposed for mouse PRDM9 [97]. It is also possible that they have no function in hotspot activity, but play a role in the regulation of gene expression as some members of the PRDM protein family do [98].

The differential retention of *α1* paralogs between salmonid genera suggests that 2 functional *Prdm9α1* copies have coexisted in the common ancestor to *Salmo* and the (*Coregonus, Thymallus, Oncorhynchus,* and *Salvelinus*) group (Fig 1). This might also be the case in primates where the pair of paralogs formed by *Prdm7* and *Prdm9* shares orthology with one ancestral copy in rodents [99]. It has been shown that changes in *Prdm9* gene dosage affect fertility in mice [100,101], suggesting that PRDM9 protein level may be limiting in some contexts. Theoretical models also predict that the loss of fitness induced by the erosion of PRDM9 targets could be compensated by increased gene dosage [47,50]. Thus, the duplication of a *Prdm9* allele might be temporarily advantageous when the amount of its target motifs starts to become too low in the genome. However, this benefit is expected to be only transient. This could explain why most (11/12) of the salmonid genomes analyzed contained a single full-length, non-pseudogenized copy of *Prdm9α1*. The succession of duplications and losses reported here in salmonids and previously described in mammals contributes to the apparent instability of *Prdm9* at the macro-evolutionary timescale.

## The reinforced PRDM9 paradox

This study uncovers a remarkable similarity in the recombination landscape regulation between salmonids and mammals. The main conclusion is that the function of PRDM9 in specifying recombination sites most likely existed in the common ancestor to vertebrates, and might be even older. Certainly, it is not a mammalian oddity. Impressively, the ultra-fast Red Queen-driven evolution of *Prdm9* and its binding motifs has been around for more than 400 My, in several vertebrate lineages [93]. This implies many thousands of amino acid substitutions per site in the ZF array [93]. Our results highlight the many open questions about this remarkable system, particularly the question of its long-term maintenance, which is now demonstrated. *Prdm9* can evidently be lost, for instance in birds and canids. Its continuous presence in most mammals, snakes, salmonids, and presumably many other taxa might be partly explained by the molecular mechanisms of PRDM9-dependent and PRDM9-independent recombination. The net output of these 2 processes is the same: CO formation. However, there may be differences in the kinetics or efficiency of DNA DSB formation and repair and thus in the robustness of CO control. This is suggested by the PRDM9-dependent recruitment of ZCWPW1, a protein that facilitates DNA DSB repair [102–104], and by the coevolution of *Prdm9* with other genes involved in DNA DSB repair and CO formation, such as *Zcwpw2, Tex15,* and *Fbxo47* [56]. Of note, *Zcwpw1, Zcwpw2,* and *Tex15* are present and intact in the 3 species that contain a full-length *Prdm9* (*S. salar, O. mykiss,* and *O. kisutch*), but are absent from the genome of *D. labrax* (S6 Table). If PRDM9 activity is linked to other molecular processes, its loss without loss of fertility may require several mutational events. Interestingly, an intermediate context, suggesting a reduction of PRDM9 activity, has been observed in the corn snake *Pantherophis guttatus*. Specifically, Hoge and colleagues [59] reported elevated recombination rates at PRDM9 binding sites and promoter-like features, introducing the idea of a "tug of war" between *Prdm9* and the default, *Prdm9*-independent, system. A recent study in mammals [105] also showed that many species with *Prdm9* make substantial use of default sites, unlike humans and mice. The relative efficiency of the *Prdm9*-independent and *Prdm9*-dependent pathways presumably evolves and differs among species. When the *Prdm9*-

independent pathway is sufficiently efficient, the conditions might be met for losing *Prdm9* irreversibly. The characterization of recombination patterns and mechanisms in species with and without *Prdm9* should help to understand the paradox of its peculiar evolution.

## Material and methods

### Ethics statement

The *S. salar* samples were collected by the Unité Expérimentale d'Ecologie et d' Ecotoxicologie Aquatique (U3E, INRAE, https://doi.org/10.15454/1.5573930653786494E12) with the authorization from an ethical committee number APAFIS#4025–201602051204637 v3. These samples were provided by the Biological Resource Centre Colisa (DOI: Biological Resource Centre Colisa) part of BRC4Env (DOI: https://doi.org/10.15454/TRBJTB), of the Research Infrastructure AgroBRC-RARe. The *O. mykiss* samples were collected in accordance with the CNRS guidelines for animal welfare and ethical authorization n° APAFIS#13616–2018021315504139 v5 issued by the local committee for ethical animal experimentation and the French ministries of research and agriculture.

### Phylogenetic analysis of PRDM9 paralogs in salmonids

We investigated the presence of full-length PRDM9 in 12 species from the 3 salmonid subfamilies (*Coregoninae*, *Thymallinae*, and *Salmoninae*). We searched for *Prdm9*-related genes by homology using the full-length copy of *O. kisutch* (coho salmon), focusing on the 3 PRDM9 canonical domains: KRAB (encoded by 2 exons), SSXRD (1 exon), and SET (3 exons). We obtained coho salmon PRDM9 from a nearly full-length coding sequence annotated in the RefSeq database (XP_020359152.1), complemented in its 3′ end using a cDNA identified in a brain RNA-seq data set sequenced with PacBio long reads (SRR10185924.264665.1). We used this reference sequence to identify, with BLAST, *Prdm9* homologs in the whole genome assembly of lake whitefish (*Coregonus clupeaformis*), European grayling (*Thymallus thymallus*), huchen (*Hucho hucho*), coho salmon (*O. kisutch*), rainbow trout (*O. mykiss*), chinook salmon (*Oncorhynchus tschawytscha*), chum salmon (*Oncorhynchus keta*), red salmon (*Oncorhynchus nerka*), pink salmon (*Oncorhynchus gorbuscha*), Atlantic salmon (*S. salar*), brown trout (*Salmo trutta*), and lake trout (*Salvelinus namaycush*), and also of northern pike (*Esox lucius*, Esocidae), a closely related outgroup, and sea bass (*Dicentrarchus labrax*). As we obtained multiple hits, we filtered out copies containing only one of the 6 exons. We compared candidates to all *PRDM*-related genes annotated in the human and mouse genomes in Ensembl to exclude non-*Prdm9* homologs. We aligned the retained exons separately using Macse (v2.06) [106] to take into account potential frameshifts and stop codons. We manually examined and edited the alignments before concatenating exons of the same copy using AMAS concat [107]. Several paralogous copies of *Prdm9* are expected to result from the 2 WGD events that occurred in the common ancestor of teleosts (Ts3R, *c.a.* 320 Mya) and salmonids (Ss4R, *c.a.* 90 Mya), respectively [63–65]. We used the location of these paralogs on pairs of ohnologous chromosomes resulting from the most recent Ss4R duplication to trace the evolutionary history of *Prdm9* duplications, retention and losses. We built the maximum-likelihood phylogeny of the 3 canonical domains using IQ-TREE [108] based on amino acid alignments, using ultrafast bootstrap with 1,000 replicates. Lastly, to identify functional *Prdm9* copies with sequence orthology to the 10 exons found in human and mouse *Prdm9* [109], we predicted the structure of each gene copy surrounded by 10 kb flanking regions using Genewise (v2.4.1) [110]. We selected representative paralogous sequences across the obtained *Prdm9* phylogenetic tree to perform a sequence similarity-based annotation of the copies in each species. See details in Supporting information (S1 Methods).

## Analysis of PRDM9 ZF diversity in rainbow trout and Atlantic salmon

We characterized the allelic diversity of the ZF domain of *Prdm9α* copies in 2 species with different functional α-paralogs: Atlantic salmon (*S. salar*) and rainbow trout (*O. mykiss*). We focused on *Prdm9α* because a previous study showed that in teleost fish, *Prdm9β* copies lack the KRAB and SSXRD domains, have a slowly evolving ZF domain, and carry a presumably inactive SET domain [19].

First, to validate the presence of expressed *Prdm9α* copies, we inferred the expression levels of multiple *Prdm9α* paralogs in immature testes from the *Salmo* and *Oncorhynchus* genera, using publicly available RNA-seq data from the Sequence Read Archive (SRA) repository. Specifically, we analyzed data from 2 *S. salar* samples (SRR1422872 and SRR9593306), 2 *O. kisutch* samples (SRR8177981 and SRR2157188), and 1 sample in *O. mykiss* (SRR5657606). Our analysis revealed high expression of 2 distinct *Prdm9α* paralogs in both genera that were previously identified in the phylogenetic analysis. We then sequenced the *Prdm9α* paralogs *α1.a.2* (full length, chromosome 5, n = 26) and *α2.2* (partial, chromosome 17, n = 20) in *S. salar*, and the *Prdm9α* paralogs *α1.a.1* (full length, chromosome 31, n = 23) and *α2.2* (full length, chromosome 7, n = 20) in *O. mykiss*.

We used wild Atlantic salmon samples from Normandy (France) and rainbow trout samples from an INRAE selected strain (S7 Table). We extracted genomic DNA from fin clips stored in ethanol at −20˚C, using the Qiagen DNAeasy Kit following the manufacturer's instructions. We measured DNA concentration and purity with a Nanodrop-1000 Spectrophotometer (Thermo Fisher Scientific) and assessed DNA quality by agarose gel electrophoresis. We designed primers using NCBI Primer Blast, ensuring specificity against the reference assemblies. Primers targeted the ZF sequence encoded in the last exon of the gene, framed by the flanking arms of the array, avoiding any specificity of the paralogous loci (S8 Table). We carried out PCR reactions using 1X Phusion HF buffer, 200 µm dNTPs, 0.5 µm forward primer, 0.5 µm reverse primer, 3% DMSO, 2.5 to 10 ng template, and 0.5 units of Phusion Polymerase (NEB) (total volume: 25 µl). Cycling conditions were: initial denaturation at 98˚C for 2 min followed by 35 cycles of 98˚C for 10 s, 66 to 70˚C for 30 s, 72˚C for 90 s, and a final elongation step at 72˚C for 3 min, followed by hold at 10˚C, in a C1000 Cycler (Bio-Rad). We examined PCR products on agarose gels and purified them using the NucleoSpin Gel and PCR clean-up kit (Machery-Nagel). We performed Sanger sequencing of single-size amplicons. Conversely, we separated by electrophoresis heterozygous samples showing 2 different length alleles, followed by cloning using the TOPO Blunt Cloning Kit (Invitrogen) and sequencing. Sequencing was done by Azenta-GeneWiz (Leipzig, Germany).

We assembled and aligned forward and reverse reads to the reference ZF array from *S. salar* ICSASG_v2 and *O. mykiss* USDA_OmykA_1.1, using SnapGene (v5.1.4.1–5.2.3). We translated contigs into amino acid sequences used to categorize individual *Prdm9α* alleles. We annotated all ZF arrays to match the C2H2 ZF motif X7-CXXC-X12-HXXXH. We reported new alleles every time we found a single amino acid variation. We aligned the DNA sequences for each allele to create a consensus sequence. We then followed [56] and [19] to compare amino acid diversity at DNA-binding residues of the ZF array (positions −1, 2, 3, and 6 of the α-helix) with diversity values at each site of the ZF array. We calculated the proportion of the total amino acid diversity (r) at DNA-binding sites as the sum of diversity at DNA-binding residues over the sum of diversity at all 28 residues of the array (see details in S1 Methods).

## Identification of DSB hotspots in rainbow trout using ChIP-sequencing

We investigated the genome-wide distribution of DMC1-bound ssDNA in *O. mykiss* testes by ChIP followed by ssDNA enrichment (DMC1-Single Strand DNA Sequencing, DMC1-SSDS). We chose 3 rainbow trout individuals from the pool of samples previously used to characterize PRDM9 ZF diversity. We determined the stage of gonadal maturation by macroscopic (whole gonads) and histological (gonad sections) analyses, according to [111] (S26 Fig and S1 Methods). As DMC1 binds to chromatin during the early stages of the meiotic prophase I, we used testes at stages III and IV from 3 individuals with different *Prdm9* genotypes (TAC-1: $Prdm9^{1/5}$, stage III; TAC-3: $Prdm9^{2/6}$, stage III; and RT-52: $Prdm9^{1/2}$, stage IV). This allowed us to compare DSB hotspots between individuals sharing or not a *Prdm9* allele.

For H3K4me3 and H3K36me3 ChIP experiments, we used the protocols described in [112,113] with some adjustments and rabbit anti-H3K4me3 (Abcam, ab8580) and anti-H3K36me3 (Diagenode, Premium, C15410192) antibodies. For DMC1 ChIP, we used previously described methods [69,114] and antibodies against DMC1. These antibodies were raised by immunization of a rabbit and a guinea pig with a His-tagged recombinant zebrafish Dmc1 (see details in S1 Methods). All ChIP experiments were performed in duplicate. A list of the samples and antibodies used for the ChIP-seq experiments, the number of mapped reads and accession numbers are in S9 Table.

For H3K4me3 and H3K36me3 ChIP-seq, we generated libraries using the NEBNext Ultra II protocol for Illumina (NEB, E7645S-E7103S), with minor adjustments. For DMC1-SSDS, we generated libraries following the Illumina TruSeq protocol (Illumina, IP-202-9001DOC), with the introduction of an additional step of kinetic enrichment, as previously described [69,114]. Libraries were sequenced on a NovaSeq6000 platform (Illumina) with S4 flow cells by Novogene Europe (Cambridge, United Kingdom).

We analyzed histone modifications with the nf-core/chipseq v1.2.1 pipeline developed by [115]. Briefly, we aligned the sequencing reads of all ChIP-seq experiments to the USDA_O-mykA_1.1 assembly with BWA (v0.7.17-r1188). For both H3K4me3 and H3K36me3 modifications, we normalized the signal based on the read coverage and by subtracting the input. We performed peak calling with MACS2 (v2.2.7.1) for both replicates and provided an input for each sample. We assessed the histone modification enrichment at DMC1 peaks, and the enrichment of H3K36me3 signal at H3K4me3 peaks in brain using the deepTools suite [116] and the bed files produced by the AQUA-FAANG project (https://www.aqua-faang.eu/). We analyzed the DMC1-SSDS data as described in [69], with some implementations described in [117], using the hotSSDS pipeline (version 1.0). We mapped reads with the modified BWA algorithm (BWA *Right Align*), developed to align and recover ssDNA fragments, as described by [114]. We normalized the signal based on the library size and the type 1 ssDNA fragments. We performed peak calling with MACS2 (v2.2.7.1) and relaxed conditions for each of the 2 replicates and provided an input control. We carried out an irreproducible discovery rate (IDR) analysis to identify reproducible enriched regions. Then, we used these peaks as DSB hotspots (see details in S1 Methods). We used the final peaks to check the distribution of ssDNA type 1 signal at DSB hotspots.

We explored the relationship between H3K4me3 and H3K36me3 signal distribution by calculating the correlation between H3K4me3 and H3K36me3 read enrichment at DSB hotspots in the RT-52 sample, of the H3K4me3 read enrichment between the TAC-1 and TAC-3 samples at the RT-52 DSB hotspots, and of the H3K36me3 read enrichment between the TAC-1 and TAC-3 samples at RT-52 DSB hotspots. Lastly, we assessed the proportion of DSB hotspots overlapping with H3K4me3 and H3K36me3 peaks.

## Reconstruction of population recombination landscapes in 3 salmonid species

**Whole-genome resequencing data.** To reconstruct population-based recombination landscapes, we collected high coverage whole-genome resequencing data from 5 natural populations of 3 salmonid species from the SRA database: coho salmon (*O. kisutch*), rainbow trout (*O. mykiss*), and Atlantic salmon (*S. salar*). We used ~20 individuals per population as recommended [83]. We retrieved 20 genomes of the Southern British Columbia population of coho salmon [118], 22 genomes of rainbow trout from North West America [119], and 60 genomes of 3 populations of Atlantic salmon belonging to the 2 major lineages from North America and Europe [120]: 20 from the Gaspesie Peninsula in Canada (GP population thereafter), 20 from the North Sea (NS population), and 20 from the Barents Sea in Norway (BS population). Sample accession numbers and locations are in S10 Table and S27 Fig.

**Variant calling.** Variants and genotypes called by [118] using GATK were used for *O. kisutch*. We followed the same methodology for variant calling and genotyping in *O. mykiss* and *S. salar*, using the GATK best-practice pipeline (> v3.8–0, see S11 Table for the detailed versions of the programs [121,122]). First, we aligned paired-end reads to their reference genome (Okis_V1, GCF_002021735.1; Omyk_1.0, GCF_002163495.1; Ssal_v3.1, GCF_905237065.1, see S12 Table for assembly statistics) using BWA-MEM (v0.7.17, Li and Durbin, 2009; -*M* option), yielding an average read coverage depth per sample of 29.54×, 24.87×, and 9.97× for *O. kisutch*, *O. mykiss*, and *S. salar*, respectively (S4 Table). We used Picard (> v2.18.29) to mark PCR duplicates and add read groups. Then, we performed variant calling separately for each individual using HaplotypeCaller before joint genotyping with GenotypeGVCFs. In total, we analyzed 9,590,270, 39,601,311, and 27,061,466 single-nucleotide polymorphisms (SNPs) for *O. kisutch*, *O. mykiss*, and *S. salar*, respectively.

After genotyping, we removed variants within 5 bp of an indel with the Bcftools filter (v 1.9; Li, 2011; -*g 5*). We filtered low-quality SNPs with Vcftools (> v 0.1.16) [123], keeping only biallelic SNPs, and excluding genotypes with low-quality scores (—*minGP 20*) and SNPs with >10% of missing genotypes (—*max-missing 0.9*). For the *S. salar* data set, we set the missingness threshold at 50% to take into account the lower sequencing coverage depth in this species. To remove the effect of poorly sequenced and duplicated regions, we kept only sites with a mean coverage depth within the 5% to 95% quantiles of that species distribution. To further eliminate shared excesses of heterozygosity due to residual tetrasomy or contaminations, we applied a Hardy–Weinberg equilibrium filter with a *p*-value exclusion threshold of 0.01 (—*hwe 0.01*). We removed singletons by applying a minor allele count (MAC) filter with Vcftools (—*mac 2*). For *S. salar*, we used the missingness, Hardy–Weinberg, and MAC filters separately for each of the 3 populations. After these filtering steps, we retrieved a total of 7,205,269, 16,079,097, and 5,575,430 SNPs for *O. kisutch*, *O. mykiss*, and *S. salar*, respectively (S4 Table).

**Variant phasing and orientation.** We used the read-based phasing approach in WhatsHap (> v0.18) [124] to identify phase blocks from paired-end reads that overlapped with neighboring individual heterozygous positions. This allowed us to locally resolve the physical phase of 73.45%, 76.98%, and 7.32% of variants for *O. kisutch*, *O. mykiss*, and *S. salar*, respectively. Then, we performed the statistical phasing of pre-phased blocks with SHAPEIT4 (> v4.2.1, [125], default settings) in each species, assuming a uniform recombination rate of 3 cM/Mb (representative of the average recombination rates in teleosts, [11]) and using the effective population size estimated from the mean nucleotide diversity of each chromosome calculated with Vcftools.

We inferred ancestral allelic state probabilities for the set of retained variants of each species with the maximum-likelihood method implemented in est-sfs (v2.04, Kimura-2-parameter

substitution model) [126], using 3 outgroups per species chosen among the available salmonid reference genomes (see details in S1 Methods). The method uses the ingroup allele frequencies and the allelic states of the outgroups to infer ancestral allelic state, taking into account the phylogenetic relationships between ingroup and outgroup species [127].

**Estimation of linkage disequilibrium (LD)-based recombination rates.** For each of the 5 population data sets (*O. kisutch*, *O. mykiss*, and *S. salar* GP, BS and NS populations, S27 Fig), we estimated the population-scaled recombination rate parameter ρ ($\rho = 4N_e r$, where $N_e$ is the effective population size and $r$ the recombination rate in M/bp) using LDhelmet (v1.19) [13]. LDhelmet relies on a reversible-jump Markov Chain Monte Carlo algorithm to infer the ρ value between every pair of consecutive SNPs. Variant orientation was provided using the probabilities, estimated by est-sfs, that the major and minor alleles were ancestral, and a transition matrix was computed following [13]. We run LDhelmet 5 times independently for each population. For each chromosome, we created the haplotype configuration files with the find_conf function using the recommended window size of 50 SNPs. We created the likelihood look-up tables once for the 5 runs with the table_gen function using the recommended grid for the population recombination rate ($\rho$/pb) (i.e., $\rho$ from 0 to 10 by increments of 0.1, then from 10 to 100 by increments of 1) and with the Watterson $\theta = 4N_e\mu$ parameter of the corresponding chromosome obtained using $\mu = 10^{-8}$. We created the Padé files using 11 Padé coefficients as recommended. We run the Monte Carlo Markov chain for 1 million iterations with a burn-in period of 100,000 and a window size of 50 SNPs, using a block penalty of 5. We checked the convergence of the 5 independent runs by comparing the estimated recombination values with the Spearman's rank correlation test (Spearman's rho >0.96; S28 Fig). We averaged and smoothened the 5 runs within 2 kb, 100 kb, and 1 Mb windows using custom python scripts.

We reconstructed the fine-scale recombination landscape of the European sea bass (*D. labrax*) to compare recombination features in salmonids with a species that lacks a complete *Prdm9* gene due to loss of the KRAB domain. We used whole-genome haplotype data obtained by phasing-by-transmission and statistical phasing [128] to infer recombination in the Atlantic sea bass population with a similar strategy, using the seabass_V1.0 genome assembly (GenBank accession number GCA_000689215.1) (S4 and S11 Tables).

We estimated $N_e$ based on the nucleotide diversity ($\theta$) measured in our population resequencing data sets, and on values of mutation rates ($N_e = \theta/4\mu$). $\theta$ was calculated on the filtered data set (keeping singletons) with Vcftools in windows of 100 kb and corrected by the proportion of SNPs discarded after filtering to account for the proportion of the genome not considered by Vcftools to estimate $\theta$. We used values of $\mu$ reported in fish: stickleback ($4.56 \times 10^{-9}$; [129]), Atlantic herring ($2.0 \times 10^{-9}$; [130]), cichlid fish ($3.5 \times 10^{-9}$; [131]), guppy ($3.44 \times 10^{-9}$; [132]), and in human ($1 \times 10^{-8}$; [133]) to obtain a range of $N_e$ estimates. The range of $\mu/r$ ratio was computed in each species, based on genome-wide recombination rate ($r$) measured from published pedigree-based genetic maps [67,75–77].

We also controlled that our estimates of recombination rates were robust to limited sequencing coverage (see section "Controlling for a possible effect of limited sequencing coverage" in S1 Methods, S29 and S30 Figs).

**Identification of LD-based recombination hotspots.** We identified recombination hotspots from the raw recombination map inferred by LDhelmet (i.e., raw hotspots) and from the 2 kb smoothed recombination map (i.e., 2 kb hotspots) using a sliding window approach. We defined hotspots as intervals between consecutive SNPs or 2 kb windows with a relative recombination rate ≥5-fold higher than the mean recombination rate in the 50 kb flanking regions. When consecutive 2 kb windows exceeded the threshold, we retained only the window with the highest rate.

## Analysis of recombination landscapes

**Comparison between DMC1 and LD-based recombination maps.** We compared DSB hotspots mapped by DMC1-SSDS of the 3 pooled samples and the LD-based recombination hotspots retrieved from the recombination landscapes of *O. mykiss*. For this, we converted the genomic positions of the DSB hotspots mapped on the OmykA_1.1 assembly to the Omyk_1.0 coordinates on which we built the LD map using the Remap program from NCBI. We compared the locations of LD-hotspots and of DSB hotspots using Bedtools intersect [134].

**Recombination at genomic features.** We investigated how DSB-based (*O. mykiss*) and LD-based (3 salmonid species and sea bass) recombination rates and hotspots were distributed relative to genomic features. We first retrieved the positions of genes, exons, and introns from genome annotations in each species. We de novo identified transposable element (TE) families in each genome using RepeatModeler (v2.0.3; option -LTRStruct) [135], before mapping TEs and low complexity DNA sequences with RepeatMasker (version 4.1.3, http://www.repeatmasker.org/, options -xsmall, -nolow). We deduced intergenic regions from gene and TE locations using Bedtools *subtract*. We defined the TSS and TES as the first and last position of a gene, respectively. For each reference genome, we predicted CGIs with EMBOSS *cpgplot* (v6.6.0) [136], using the parameters *-window 500 -minlen 250 -minoe 0.6 -minpc 0*. It should be noted that the criteria that are classically used to predict CGIs in mammals and birds (CpG observed/expected ratio >0.6, GC content >50%) are not appropriate for teleost fish in which CGIs are CpG-rich but have a low GC content [78,79]. Therefore, we predicted CGIs based only on their CpG content, without any constraint on their GC content. We confirmed that these criteria efficiently predicted TSS-associated CGIs, using whole genome DNA methylation and H3K4me3 data from rainbow trout and coho salmon (see S1 Analysis).

We investigated DSB hotspots overlap with genomic features and their distance to the nearest promoter-like feature (TSS) using Bedtools. For this, we analyzed DNA DSB distribution using DMC1-SSDS read enrichment as metric. As the coordinates of genomic features were mapped on Omyk_1.0, we converted them to the OmykA_1.1 assembly using the NCBI Remap tool.

We assessed population recombination rate (2 kb scale) variations in function of the distance to the nearest TSS (overlapping or not with a CGI) using the *distanceToNearest* function of the R package *GenomicRanges* [137]. We retrieved the averaged recombination rates at each genomic feature (i.e., genes, exons, introns, intergenic regions, TSSs within and outside CGIs and TEs) using the *subsetByOverlaps* function of the same package. We compared recombination rates at genomic features in the 5 salmonid populations and in sea bass.

Lastly, we investigated the effect of SNP density, GC content, and TEs on the population recombination rate variation and the presence of recombination hotspots. We calculated SNP density, TE density, and GC content in non-overlapping 100 kb windows and compared them with the window-averaged recombination rates using the Spearman's rank correlation test. We assessed the association of hotspots with SNP density and GC content at the 2 kb scale.

**Comparison of LD-based landscapes between populations and species.** We assessed the correlation between the 100 kb smoothed recombination maps of each of the 3 *S. salar* populations using the Spearman's rank test. We identified shared hotspots between populations as overlapping 2 kb hotspots using Bedtools *intersect*. To compare the recombination hotspots of the *O. kisutch* and *O. mykiss* populations, we used a reciprocal blast approach to identify homologous regions of the genome in these 2 species (see S1 Methods). We used random permutations to calculate the expected amount of hotspot overlap between pairs of the 3 *S. salar* populations and between the 2 *Oncorhynchus* populations. We drew random spots (same

number as that of the 2 kb hotspots) 100 times from the genome for each population using Bedtools *shuffle*, after applying a genome mask to discard the regions with a nucleotide diversity lower and higher than the 2.5 and 97.5th quantile, the 0.1% highest recombination rate values, the 10% larger gap sizes, and genuine hotspots, to control for diversity level, extreme $\rho$ values, and genome gaps. We compared each of these random sets to those of the compared population to calculate the average overlap expected only by chance.

**Identification of DNA motifs at hotspots and motif erosion.** In rainbow trout, we performed motif detection analysis at DSB hotspots using the MEME Suite [138], focusing on the RT-52 data set due to its high number of DSB hotspots (DMC1 peaks). We defined 2 subsets of allele-specific hotspots using Bedtools *intersect*: Allele 1 set [RT-52 DMC1 peaks (center ± 200 bp) overlapping with H3K4me3 and H3K36me3 peaks from TAC-1 ($N$ = 300)] and Allele 2 set [RT-52 DMC1 peaks (center ± 200 bp) overlapping with H3K4me3 and H3K36me3 peaks from TAC-3 ($N$ = 254)]. We used MEME-ChIP [139] to detect motifs. Then, we assessed motif enrichment in DSB hotspots relative to control sequences using FIMO [140] and evaluated central enrichment using CentriMo [141]. We also quantified the fold-enrichment of the detected motifs at LD-based hotspots (see S1 Methods).

In Atlantic salmon, we used STREME from the MEME Suite [142] to find motifs between 10 and 20 bp in length that were enriched at LD-based hotspot positions, compared with control random sequences with a similar GC-content distribution. To obtain these controls, we randomly drew windows (totaling 10 times the number of hotspots from the reference genome), and we applied the same genome mask as in the previous section to select random spots, controlling for diversity, high recombination rates and genome gaps and to exclude hotspots. To select a subset of controls matching the GC-content of hotspots, we binned the hotspot GC-content distribution (bin width = 0.025) and then drew the same number of random spots for each GC-content bin. We retained the detected motifs as potential PRDM9-binding motifs if they were enriched ≥2-fold at the hotspots compared with the control sequences and were found in ≥5% of hotspots. We searched for motifs associated with the hotspots of each *S. salar* population and with the shared hotspots between pairs of populations.

We then tested whether the candidate motifs showed signs of erosion between lineages by comparing the number of motifs present in the available long-read genome assemblies from 5 North American and 7 European Atlantic salmon genomes. To take into account potential differences in assembly size, we aligned these 12 genomes with SibeliaZ [143] and retrieved collinear blocks that represented 89.5% of the whole genome alignment. We then used FIMO [140] to count motif occurrence in the aligned fraction of each genome, using a *p*-value cut-off of 1.0E-7. To assess the statistical significance of motif erosion in a given lineage, we obtained a null distribution of the between-lineage difference in motif occurrence by running FIMO on 100 random permutations of the candidate motif matrix.

## Supporting information

**S1 Methods. Molecular, genomic, and population genetics methods.**
(DOCX)

**S1 Analysis. Prediction of CGI-associated TSSs in salmonids.** Fig A. Relationship between base composition and DNA methylation level in promoter regions of coho salmon (*Oncorhynchus kisutch*). Fig B. Relationship between DNA methylation level in the promoter regions of coho salmon and the presence of a nearby *fpCGI*. Fig C. Relationship between base composition and H3K4me3 in promoter regions of the rainbow trout (*Oncorhynchus mykiss*). Fig D. Relationship between H3K4me3 in the promoter regions of the rainbow trout and the

presence of a nearby *fpCGI*.
(DOCX)

**S1 Table. Chromosome location of the retained PRDM9 paralog copies.** The index allows to identify the corresponding copy in the phylogeny of the *α* paralog copies in Fig 1A and the β copies in S1 Fig. We retrieved the location of the regions covering the 3 domains KRAB, SSXRD, and SET obtained from the blast analysis. The start and end positions correspond to the start position of the first exon blasted and the end position of the last exon blasted.
(DOCX)

**S2 Table. Fine scale variations in recombination rates and raw recombination hotspots.** Summary statistics of the variations in recombination rates smoothed in 2 kb sliding windows, and of recombination hotspots retrieved from the inter-SNP recombination landscapes. The raw hotspots were defined as the consecutives inter-SNP windows with a recombination rate 5-fold higher than the 50 kb flanking regions.
(DOCX)

**S3 Table. Overlaps between DSB hotspots and TSS/TES regions.** Overlaps were assessed for 400-bp wide windows centered on DSB hotspot centers and TSS/TES regions, defined as sequences found within 1 kb of distance from the transcription start/end site. The expected overlaps were estimated as the chance for 2 kb windows of overlapping 400 bp DSB hotspots genome wide.
(DOCX)

**S4 Table. Summary statistics of the LD-landscape reconstruction pipeline.** Information is retrieved on the reference genome sizes, population sample sizes, mapping depth statistics, variant calling statistics, and effective sizes of genomic features for each population.
(DOCX)

**S5 Table. Comparison of fine-scale recombination landscapes across vertebrates.**
(DOCX)

**S6 Table. Presence of *Zcwpw1*, *Zcwpw2*, *Tex15*, and *Fbox47* genes in the genomes of analyzed species.**
(DOCX)

**S7 Table. List of genotyped samples.** Details about the *Salmo salar* and *Oncorhynchus mykiss* samples used in this study and the corresponding *Prdm9α* genotypes identified.
(DOCX)

**S8 Table. List of primers.** The primers used in this study to genotype the zinc finger array of *Prdm9α* in *Salmo salar* and *Oncorhynchus mykiss*.
(DOCX)

**S9 Table. List of ChIP-seq experiments performed.** Details about the ChIP-seq experiment performed in *Oncorhynchus mykiss* testes. In the third column DMC1-R1 or -GP refer to the animal used to raise the antibody, respectively, rabbit individual 1 and guinea pig.
(DOCX)

**S10 Table. Sample accession number and location.** Population samples of *O. kisutch*, *O. mykiss*, and *S. salar* used to build the linkage disequilibrium-based recombination landscapes were selected from [118–120].
(DOCX)

**S11 Table. Program versions.** Details of the program versions used at each step of the reconstruction of LD-based recombination landscapes in the 5 salmonid populations. * The *D. labrax* data set was taken from [128], who used a reference panel of 22 genomes fully phased-by-transmission using trio-sequencing as a learning reference for the statistical phasing of 46 additional genomes with Eagle2 v2.4. Variants were oriented using whole-genome resequencing data (>20×) from the closely related species *Dicentrarchus punctatus*, which was used as an outgroup.
(DOCX)

**S12 Table. Assembly statistics of the reference genome of *O. kisutch*, *O. mykiss*, and *S. salar* that were used to map the population resequencing data for the LD-based recombination landscapes and the ChIP-Seq DMC1 peaks.**
(DOCX)

**S13 Table. Mean sequence identity score obtained from the blast search of the 100 kb flanking sequences of the variants in the ingroup species against the reference genome of each outgroup.** Outgroups 1, 2, and 3 for *O. kisutch* were *O. tshawytscha*, *O. nerka*, and *O. mykiss*, respectively; *O. tschawytscha*, *O. nerka*, and *O. kisutch* for *O. mykiss*; and *Salmo trutta*, *Salvelinus alpinus*, and *O. mykiss* for *S. salar*.
(DOCX)

**S1 Fig. Phylogenetic distribution of PRDM9β paralogs in 12 salmonids, the northern pike (*Esox lucius*) and the European sea bass (*Dicentrarchus labrax*) as outgroup species.** The phylogenetic tree was realized on the concatenated 3 exons of the SET domain, with 1,000 bootstrap replicates (values shown at nodes). To facilitate visualization, the branch of *D. labrax* is not drawn to scale. In column from left to right, (i) species; (ii) annotated paralog copy; (iii) Prdm9 copy status. The scale bar is in unit of substitution per site. The right panel shows the coding potential of each paralog and indicates the presence of substitutions in the catalytic tyrosines of the SET domain (Y276, Y341, and Y357). Canonical (full length) PRDM9 proteins contain 4 key domains: KRAB (encoded by 2 exons), SSXRD (encoded by 1 exon), SET (encoded by 3 exons), and the ZF array (encoded by 1 exon). Complete exons are shown in blue. Missing or truncated exons are shown in pink. Other regions of the protein (upstream of the KRAB domain, and between KRAB and SSXRD), are encoded by additional exons (not shown here), that are not conserved between α and β clades. All β copies have lost KRAB and SSXRD domains, and have substitutions in at least two of the 3 catalytic tyrosines of the SET domain. β copies are well conserved across all species (including in the ZF array), which indicates that these truncated PRDM9 homologs are under purifying selection, and hence that they have a function. The last column indicates indexes referring to the S1 Table with additional information on the corresponding copy. The data and codes underlying this figure can be found in https://doi.org/10.5281/zenodo.11083953.
(DOCX)

**S2 Fig. Chromosome position of the tandem duplicated PRDM9α a and b copies.** Relative position and orientation of the **a** (in red) and **b** (in blue) tandem duplicated copies of the PRDM9α1.1 and 1.2 paralogs for each species. The chromosome/scaffold name on which the copy seats is shown. α1.1 and 1.2, which occur as single copies in some species, are also shown (in gray). The data underlying this figure can be found in S1 Table.
(DOCX)

**S3 Fig. Amino acid diversity in full-length and partial PRDM9 zinc fingers in *S. salar* and *O. mykiss*.** Amino acid sequences of all unique zinc fingers found in alleles identified in

*S. salar* PRDM9α1.a.2 and α2.2, and in *O. mykiss* PRDM9α1.a.1 and α2.2 (Figs 2A and S5). In bold colored boxes are indicated the 3 hypervariable DNA-binding residues. In red are reported the cysteine (C) and histidine (H) residues involved stabilizing the structure of the array. In blue are indicated the polymorphic residues compared to the consensus, outside the 3 amino acids in contact with DNA. In shaded gray are reported the synonym variations in respect to the consensus. The complementary information about the DNA sequences of all alleles identified is available in S1 Methods.
(DOCX)

**S4 Fig. Graphical view of PRDM9 paralogs.** Cartoon showing the functional domains of PRDM9 paralogs analyzed in this study. The amino acid sequences were obtained from the reference genome and analyzed using previously described methodology [144]. α1 copies and the *O. mykiss* α2.2 copy possess a complete KRAB domain, and we refer to these copies as canonical PRDM9. *S. salar* α2.2 copy possess a partial KRAB domain, and we refer to this copy as truncated PRDM9. All 4 copies present the 3 catalytic tyrosine residues in the SET domain, required for methyltransferase activity.
(DOCX)

**S5 Fig. PRDM9α2.2 zinc finger allelic diversity in *S. salar* and *O. mykiss*.** (**A**) Structure of PRDM9 zinc finger arrays of identified alleles in *S. salar* PRDM9α2.2 and *O. mykiss* PRDM9α2.2. Colored boxes represent unique zinc fingers, characterized by the 3 amino acids in contact with DNA (3-letter code). Additional variations relative to a reference sequence are indicated between brackets. A white star indicates the zinc fingers missing one amino acid residue (27 a.a. instead of 28). The complete zinc finger amino-acid sequences are shown in S3B Fig. Frequencies of the alleles displayed on panel A among the 20 *S. salar* and 20 *O. mykiss* individuals that were genotyped for PRDM9. (**C**) Distribution of amino acid diversity among all unique zinc fingers found in alleles displayed on panel A, following previously described methodology [19]. The amino acid diversity is plotted as a function of amino acid position in the ZF alignment, ranging from position 1 to position 28 (first and last residues) of a ZF unit. The ratio of amino acid diversity at DNA-binding residues of the ZF array (−1, 2, 3, and 6), indicated as r, is shown in the upper box. The data underlying this figure can be found in https://doi.org/10.5281/zenodo.11083953 and in S7 Table.
(DOCX)

**S6 Fig. Meiotic DSB hotspots features in *O. mykiss*.** (**A**) Average profile of DMC1 ChIP-seq ssDNA fragments orientation (fragments per million, FPM) in TAC-1, TAC-3, and RT-52 testes, at DSB hotspots detected in TAC-1, TAC-3, and RT-52. The profile from each experiment performed is shown (2 replicates/sample). Signal mapped on the forward strand is depicted in blue, signal aligned to the reverse strand is shown in green, as shown in the cartoon on top of the panel. (**B**) Upset plot showing intersections between DSB hotspots from TAC-1 ($n = 616$), TAC-3 ($n = 209$), and RT-52 ($n = 1,924$). (**C**) DMC1 ChIP-seq signal fold enrichment (scaled by the average signal in intergenic regions) at multiple genomic features. TSS inside and outside CGIs are highlighted in purple and turquoise, respectively. The data and codes underlying this figure can be found in https://doi.org/10.5281/zenodo.11083953 and https://zenodo.org/records/14198863.
(DOCX)

**S7 Fig. Distribution of DSB hotspots along chromosomes in *O. mykiss*.** (**A**) Distribution of DSB hotspots from TAC-1, TAC-3, and RT-52 along chromosomes (paces of 1/30 of chromosome length). (**B**) Average profile and heatmap of DMC1 ChIP-seq ssDNA fragments orientation (fragments per million, FPM) in TAC-1, TAC-3, and RT-52 testes, at DSB hotspots

shared by pairs of samples. Shared DMC1 peaks: TAC-1 intersecting RT-52 ($n = 167$), TAC-1 intersecting TAC-3 ($n = 55$), and RT-52 intersecting TAC-3 ($n = 42$). The plots depict one replicate for each experiment performed (replicate 1). Signal mapped on the forward strand is depicted in blue, signal aligned to the reverse strand is shown in green. The data and codes underlying this figure can be found in https://doi.org/10.5281/zenodo.11083953 and https://zenodo.org/records/14198863.
(DOCX)

**S8 Fig. Histone modification signal at H3K4me3 peaks and at DSB hotspots.** (**A**) Average profile and heatmap of H3K36me3 ChIP-seq signal in TAC-1 (blue) and TAC-3 (green) testes, at H3K4me3 peaks detected in brain (Aqua-FAANG). (**B**) Average profile and heatmap of H3K4me3 (left) and H3K36me3 (right) ChIP-seq signal in TAC-1 testes, at DSB hotspots detected in TAC-1 (blue), TAC-3 (cyan), and RT-52 (yellow). (**C**) Average profile and heatmap of H3K4me3 (left) and H3K36me3 (right) ChIP-seq signal in TAC-3 testes, at DSB hotspots detected in TAC-1 (blue), TAC-3 (cyan), and RT-52 (yellow). The data underlying this figure can be found in https://doi.org/10.5281/zenodo.11083953.
(DOCX)

**S9 Fig. Correlation of H3K4me3 and H3K36me3 signal at RT-52 hotspots.** (**A**) Scatterplots showing H3K4me3 and H3K36me3 ChIP-seq signal in TAC-1 and TAC-3 testes, at RT-52 DSB hotspots. (**B**) Left panels, scatterplots representing H3K4me3 (top) or H3K36me3 (bottom) ChIP-seq signal in TAC-1 and TAC-3 testes, at RT-52 DSB hotspots. Right panels, numbers of RT-52 hotspots with H3K4me3 (top) or H3K36me3 (bottom) ChIP-seq signal under or above 1 in TAC-1 and TAC-3. Chi-square test of homogeneity. The data underlying this figure can be found in https://doi.org/10.5281/zenodo.11083953.
(DOCX)

**S10 Fig. Broad scale recombination rate variations along the genome.** Recombination rates were averaged into percentiles of chromosome length and scaled by the genomic mean for (**A**) *O. kisutch* (in orange), *O. mykiss* (in green), and *S. salar* (in blue, only the NS population is shown); and (**B**) *D. labrax* (in red) and the three-spined stickleback *Gasterosteus aculeatus* (in black, data from [61]). The data and codes underlying this figure can be found in https://doi.org/10.5281/zenodo.11083953.
(DOCX)

**S11 Fig. Fine-scale recombination landscapes of *O. kisutch* (in orange), *O. mykiss* (in green), and *S. salar* (in blue, only the NS population is shown), with recombination rates smoothed in 2 kb sliding windows.** The data and codes underlying this figure can be found in https://doi.org/10.5281/zenodo.11083953.
(DOCX)

**S12 Fig. Proportion of recombination according to proportion of the genome for *O. kisutch* (orange), *O. mykiss* (green), *S. salar* (shades of blue), and *D. labrax* (gold).** The data and codes underlying this figure can be found in https://doi.org/10.5281/zenodo.11083953.
(DOCX)

**S13 Fig. Proportion of hotspots (in %) according to raw hotspot size.** Hotspots were defined as consecutives windows of 2 adjacent SNPs in which the recombination rate is at least 5-fold higher than the 50 kb flanking regions. The data and codes underlying this figure can be found in https://doi.org/10.5281/zenodo.11083953.
(DOCX)

**S14 Fig. Comparison between the LD-based recombination landscape and the ChIP-Seq DMC1 map of the rainbow trout *O. mykiss*.** (**A**) Venn diagram showing the percentage of shared peaks between the ChIP-Seq peaks of the pooled samples (in brown) and the LD-based hotspots (in green). The percentage has been calculated using the number of DMC1 peaks as the denominator. (**B**) Random expected (blue) and observed values (orange) of shared peaks between LD and ChIP-Seq maps. (**C**) Recombination rates ρ in the syntenic location of the ChIP-Seq peaks, in the LD hotspots, in the shared ChIP-Seq and LD windows (i.e., 116 ChIP--Seq peaks shared with LD hotspots) and in the background landscapes (i.e., the genomic windows not containing neither a LD hotspot nor a ChIP-Seq peak). The data and codes underlying this figure can be found in https://doi.org/10.5281/zenodo.11083953.
(DOCX)

**S15 Fig. Broad scale variation in genomic variables according to recombination rates.** (**A**) SNP density (per kb). (**B**) GC-content. (**C**) TE density (per 100 kb). Recombination rates, SNP density GC-content, and TE density were averaged in 100 kb sliding windows. Significance *p*-value of Spearman's rank test $<0.05$ are indicated by an asterisk in panels. The vertical dashed line is the mean recombination rates and the horizontal dashed line is the mean y variable. The data and codes underlying this figure can be found in https://doi.org/10.5281/zenodo.11083953.
(DOCX)

**S16 Fig. Genetic diversity and base composition at recombination hotspots.** (**A**) SNPs density (per kb) and (**B**) GC content, according to distance to the nearest recombination hotspots. SNP density, GC-content, and recombination rates were averaged in 2 kb windows. Colored (orange, green, and blue) dashed lines show the mean of the y variable at hotspots of the corresponding populations, the black dashed line is the genomic mean (outside hotspots). Loess curves are shown for a span of 0.5. The data and codes underlying this figure can be found in https://doi.org/10.5281/zenodo.11083953.
(DOCX)

**S17 Fig. Average recombination rates in TEs families.** Tc1-mariner, a family of LTR, is shown. The data and codes underlying this figure can be found in https://doi.org/10.5281/zenodo.11083953.
(DOCX)

**S18 Fig. Inter- and intra-chromosome variation in recombination rates in residual tetra-ploid chromosomes.** (**A**) Averaged recombination rate per chromosome. Gray bars indicate chromosomes without residual tetrasomic regions (2N), and yellow bars indicate chromosomes with residual tetrasomic regions (4N) in the 5 *Oncorhynchus* and *Salmo* populations. Gray dashed lines represent the genome averaged recombination rates of the 2N chromosomes, and the yellow line in the 4N chromosomes. Recombination rates are significantly higher in 4N chromosomes compared to 2N chromosomes (Student test, t(13.258) = −3.9404, $p < 0.05$) in *O. mykiss* population, but not in *O. kisutch* (Student test, t(25.867) = −0.88786, $p > 0.05$) neither in *S. salar* populations (Student test, t(23.519) = −0.0026857, $p > 0.05$ for GP, t(20.99) = −2.2572, $p < 0.05$ for BS and t(19.677) = −1.9741, $p = 0.06258$ for NS). (**B**) Recombination rates along the genome. Recombination rates were averaged into percentiles of chromosome length and scaled by the genomic mean. Same color as panel A. The data and codes underlying this figure can be found in https://doi.org/10.5281/zenodo.11083953.
(DOCX)

**S19 Fig. Recombination rates at genomic features in residual tetraploid chromosomes.** (**A**) Fold recombination rates (scaled by the average recombination rate at 50 kb from the nearest feature) according to distance to the nearest promoter-like features (i.e., TSS overlapping or not a CGI). Recombination rates in chromosomes not containing residual tetraploid regions are shown by the continuous line and by the dashed line for the 4N chromosomes. (**B**) Fold recombination rates (scaled by the average recombination rates in intergenic regions) in genomic features, in 2N (gray) and 4N (yellow) chromosomes. The horizontal line shows the intergenic recombination level. TSS and TES were defined as the first and last positions of genes. CGIs were mapped with EMBOSS using CpGoe > 0.6 and GC > 0. The data and codes underlying this figure can be found in https://doi.org/10.5281/zenodo.11083953.
(DOCX)

**S20 Fig. Significance of hotspots sharing between closely related populations.** Random expectations (blue) and observed values (orange) of shared hotspots between (**A**) *O. kisutch* and *O. mykiss*; between *S. salar* populations (**B**) GP and BS; (**C**) GP and NS; and between (**D**) BS and NS. Shared hotspots were defined as 2 kb hotspots overlapping by at least 1 bp. Percent shared is calculated using the number of hotspots in the species/population with fewer hotspots as the denominator. The expected distribution of shared hotspots has been obtained from 1,000 pairwise comparisons of random spot. The data and codes underlying this figure can be found in https://doi.org/10.5281/zenodo.11083953.
(DOCX)

**S21 Fig. Pairwise comparison of 100 kb smoothed recombination maps between *S. salar* populations.** (**A**) Comparison between GP and BS populations. (**B**) Comparison between GP and NS populations. (**C**) Comparison between BS and NS populations. Spearman's rank test $p$-value <0.05. Loess curves are shown for a span of 0.7. The data and codes underlying this figure can be found in https://doi.org/10.5281/zenodo.11083953.
(DOCX)

**S22 Fig. DSB and LD hotspots are enriched in PRDM9 allele-specific motifs.** Frequency of sequences with at least one hit for PRDM9 allele 1 (left) and allele 2 (right) motifs at allele 1 and allele 2 sites, RT-52, TAC-1 and TAC-3 DSB hotspots, LD-hotspots and control sites. Fold enrichment relative to the control sites is shown on top of each column. The associated $p$-values indicate significant differences in fold enrichment relative to the control (Fisher exact test). "NS" indicates not significant ($p > 0.05$). The data and codes underlying this figure can be found in https://doi.org/10.5281/zenodo.11083953.
(DOCX)

**S23 Fig. DSB and LD-based hotspots are enriched in PRDM9 allele-specific motifs.** Positional distribution of hits for *Prdm9* allele 1 (pink) and allele 2 motifs (green) in RT-52, TAC-1, TAC-3 DSB hotspots, LD stronger hotspots ($n = 5,000$) and control sites ($n = 5,000$). The distribution is shown from the center of the sequence with a range of ±2.5 kb for the DSB hotspots and the control sites. The LD hotspots were centered on the SNP interval showing the highest recombination rate ($\rho$/bp) and the distribution extends up to 7.5 kb from the refined center. The signal is smoothed by weighted moving average and hits were calculated either in a 750 bp window for the LD hotpots and in a 250 bp window for all other sequences. The statistical significance of motif enrichment, adjusted for multiple tests, is shown (one-tail binomial test). "ns" indicates non-significant enrichment ($p > 0.05$). The data underlying this figure can be found in https://doi.org/10.5281/zenodo.11083953.
(DOCX)

**S24 Fig. Motifs enrichment at population-specific and shared recombination hotspots in *S. salar* populations.** (**A**) Average recombination rate in motifs found enriched at hotspot. Yellow boxes show motifs found in at least 5% of hotspots showing 2-fold enrichment compared to the control set of random spots. (**B**) Average recombination rate in hotspots containing the retained motifs from panel A (with the corresponding motifs shown) compared to hotspots not containing the retained motifs. Significant Student's tests are indicated (***, *p*-value <0.05). (**C**) Percentage of hotspots containing the retained motifs shown in yellow. The data and codes underlying this figure can be found in https://doi.org/10.5281/zenodo.11083953.
(DOCX)

**S25 Fig. Genome wide distribution of PRDM9α motifs along chromosomes in *O. mykiss*.** (**A**) Distribution of PRDM9[1] (*n* = 68,047) and PRDM9[2] (*n* = 59,986) motifs in rainbow trout genome along chromosomes (paces of 1/30 of chromosome length). (**B**) Distribution of motifs enriched in the shared hotspots between the BS and NS populations of the Atlantic salmon (*n* = 936).
(DOCX)

**S26 Fig. Histology and immunostaining of trout gonads.** (**A**) Hematoxylin-eosin-stained histological sections of testes from *O. mykiss* samples used in this study. In TAC-1, TAC-3 and RT-52 the seminiferous tubules were filled with round cells, mostly primary spermatocytes (Sc), some spermatids (ST), few spermatogonia (Sg), and almost no mature spermatozoa visible (Sz). For meiotic cells, between brackets is indicated the substage of prophase I: leptotene (L), zygotene (Z), diplotene (D), and diakinesis (DK). Scale bars are 20 μm. (**B**) Immunofluorescence of SYCP3, SMC3, and DMC1 in testes sections from a stage III *O. mykiss* sample not used for ChIP in this study. Scale bars are 10 μm.
(DOCX)

**S27 Fig. Sample location.** The 20 individuals of *O. kisutch* were samples in the Columbia River (in orange) [118], the 22 samples of *O. mykiss* come from North America rivers (in green) [119], and the 60 individuals of *S. salar* were sampled in Canada and Norway [120]. Based on population structure analysis, we subdivided the Atlantic salmon samples into 3 populations (in shades of blue): Gaspesie-Anticosti (GP), Barents sea (BS), and North sea (NS). The basemap shapefile used in this figure was derived from the CIA World DataBank II, accessed via the mapdata package in R.
(DOCX)

**S28 Fig. Pairwise correlation between the 5 independent runs of LDhelmet.** Spearman's rank correlation matrix for the 5 populations, *p*-value <0.05. The data and codes underlying this figure can be found in https://doi.org/10.5281/zenodo.11083953.
(DOCX)

**S29 Fig. Patterns of population recombination rate variation controlled for sequencing coverage.** (**A**) Average population recombination rate; (**B**) average hotspot density; and (**C**) average population recombination rate in hotspots, according to average sequencing coverage. Student *t* test *p*-values and Cohen's D coefficient are shown in A, B, and C. (**D**) Fold recombination rates (scaled by the average recombination rate at 50 kb from the nearest feature) according to the distance to the nearest TSS (overlapping or not with a CGI) shown in color, and to the mean depth (shown by the line type). (**E**) Fold recombination rates (scaled by the average recombination rates in intergenic regions); and (**F**) hotspot density at the indicated genomic features according to the mean coverage shown in color. TSS and TES were defined as the first and last positions of each gene. CGIs were mapped using EMBOSS with

CpGoe > 0.6 and GC > 0. "High" sequencing coverage corresponds to the half of the recombination map with the highest depth and "low" sequencing coverage corresponds to the half of the recombination map with the lowest depth. Only the NS population of *S. salar* is shown. The data and codes underlying this figure can be found in https://doi.org/10.5281/zenodo.11083953.
(DOCX)

**S30 Fig. Hotspot sharing between populations controlled for sequencing coverage.** Fold recombination rates around hotspots and at orthologous loci in the 2 taxa, for the 2 *Oncorhynchus* species (**A** and **B**), the American (GP population) and European (BS and NS populations) *S. salar* lineages (**D** and **E**), and between the 2 closely related European *S. salar* populations (BS and NS) (**G** and **H**), according to the mean coverage shown in panels. Random expectations (blue) and observed values (orange) of shared hotspots between (**C**) *O. kisutch* and *O. mykiss*; between *S. salar* populations (**F**) GP and BS; (**I**) BS and NS, according to the mean coverage shown in panels. Shared hotspots were defined as 2 kb hotspots overlapping by at least 1 bp. Percent shared is calculated using the number of hotspots in the population with fewer hotspots as the denominator. The expected distribution of shared hotspots has been obtained from 1,000 pairwise comparisons of random spots. "High" sequencing coverage corresponds to the half of the recombination map with the highest depth and "low" sequencing coverage corresponds to the half of the recombination map with the lowest depth. The data and codes underlying this figure can be found in https://doi.org/10.5281/zenodo.11083953.
(DOCX)

## Acknowledgments

We thank the European project AQUA-FAANG for sharing epigenetic data of *O. mykiss* and *S. salar* prior to publication. These data are now available at ENA (PRJEB57956 and PRJEB55063). We thank the Aqua Genome project for providing published Wild Atlantic salmon genome sequencing data. We particularly thank Sigbjørn Lien, Marie-Odile Baudement, and Yann Guiguen for helpful discussions, and Gareth Gillard for bioinformatic analysis of AQUA-FAANG data. We also thank Guillaume Evanno for providing wild Atlantic salmon samples, and Rajalekshmi Navaryana Sarma for help with *Prdm9* genotyping. We are grateful to Ben Coop and Eric Rondeau for providing us with the re-sequenced genomes and genotype data set of coho salmon. We thank Louis Bernatchez, Eric Normandeau, and Maeva Leitwein for providing the whole genome bisulfite sequencing methylation data of coho salmon. We also thank Thomas Brazier, Nicolas Lartillot, and Carina Mugal for helpful discussions and feedback.

## Author Contributions

**Conceptualization:** Nicolas Galtier, Frédéric Baudat, Laurent Duret, Pierre-Alexandre Gagnaire, Bernard de Massy.

**Data curation:** Marie Raynaud, Paola Sanna, Julie Clément.

**Formal analysis:** Marie Raynaud, Paola Sanna, Julie Clément, Nicolas Galtier, Frédéric Baudat, Laurent Duret, Pierre-Alexandre Gagnaire, Bernard de Massy.

**Funding acquisition:** Nicolas Galtier, Laurent Duret, Bernard de Massy.

**Investigation:** Marie Raynaud, Paola Sanna, Julien Joseph, Julie Clément, Frédéric Baudat, Laurent Duret, Pierre-Alexandre Gagnaire, Bernard de Massy.

**Methodology:** Marie Raynaud, Paola Sanna, Julien Joseph, Julie Clément, Yukiko Imai, Jean-Jacques Lareyre, Audrey Laurent, Frédéric Baudat, Laurent Duret, Pierre-Alexandre Gagnaire, Bernard de Massy.

**Project administration:** Nicolas Galtier, Laurent Duret, Bernard de Massy.

**Resources:** Yukiko Imai, Jean-Jacques Lareyre, Audrey Laurent.

**Supervision:** Nicolas Galtier, Frédéric Baudat, Laurent Duret, Pierre-Alexandre Gagnaire, Bernard de Massy.

**Validation:** Laurent Duret, Pierre-Alexandre Gagnaire, Bernard de Massy.

**Writing – original draft:** Marie Raynaud, Paola Sanna, Frédéric Baudat, Laurent Duret, Pierre-Alexandre Gagnaire, Bernard de Massy.

**Writing – review & editing:** Marie Raynaud, Paola Sanna, Frédéric Baudat, Laurent Duret, Pierre-Alexandre Gagnaire, Bernard de Massy.

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
