## [Editor Report · Decision Letter 0]

26 Apr 2024

Dear Bernard, 

Thank you for submitting your manuscript entitled "PRDM9 drives the location and rapid evolution of recombination hotspots in salmonids" for consideration as a Research Article by PLOS Biology. I'd like to apologise for the extraordinary delay incurred while we sought external advice.

Your Appeal has now been evaluated by the PLOS Biology editorial staff and I am writing to let you know that we would like to send your submission out for external peer review.

Once your full submission is complete, your paper will undergo a series of checks in preparation for peer review. After your manuscript has passed the checks it will be sent out for review. To provide the metadata for your submission, please Login to Editorial Manager (https://www.editorialmanager.com/pbiology) within two working days, i.e. by Apr 30 2024 11:59PM.

Kind regards,

Roli

Roland Roberts, PhD

Senior Editor

PLOS Biology

rroberts@plos.org

---

## [Decision Letter · Decision Letter 1]

18 Jun 2024

Dear Bernard,

Thank you for your patience while your manuscript "PRDM9 drives the location and rapid evolution of recombination hotspots in salmonids" was peer-reviewed at PLOS Biology. It has now been evaluated by the PLOS Biology editors, an Academic Editor with relevant expertise, and by four independent reviewers. Please accept my apologies for the additional delay; we had some difficulties contacting the Academic Editor. In the end we chose to send the decision letter without their input, as the way forward seemed very clear, but it is possible that the AE might make some minor requests in the next round.

Reviewer #1 thinks that the conclusions are well supported, but that "the comparison between species is hard to interpret as presented" and that several analyses need to be improved. S/he wants more information in Table 1, consideration of the effects of the low read-depth of the salmon data, use of better statistical controls, and better framing of the paper with respect to the literature (e.g. findings on snakes); there is also a list of more minor concerns. Reviewer #2 is positive but has quite a few technical and presentational requests, mostly minor, but some will need some new analyses. Reviewer #3 is very positive and simply has one discussion point. Reviewer #4 is somewhat less enthusiastic about the advance, but has a long list of requests for clarification and improvement.

In light of the reviews, which you will find at the end of this email, we would like to invite you to revise the work to thoroughly address the reviewers' reports.

Given the extent of revision needed, we cannot make a decision about publication until we have seen the revised manuscript and your response to the reviewers' comments. Your revised manuscript is likely to be sent for further evaluation by all or a subset of the reviewers.

**IMPORTANT - SUBMITTING YOUR REVISION**

*Re-submission Checklist*

*Published Peer Review*

*PLOS Data Policy*

*Blot and Gel Data Policy*

Sincerely,

Roli

Roland Roberts, PhD

Senior Editor

PLOS Biology

rroberts@plos.org

REVIEWERS' COMMENTS:

Reviewer #1:

The authors use a combination of experimental and statistical approaches to examine the role of PRDM9 and infer fine-scale recombination rates in four fish species. They provide compelling evidence for the role of PRDM9 in directing double strand break locations in fish that carry an intact version of the gene, and a comparison of recombination landscapes in fish with and without an intact version.

My impression is that the overall conclusions are well supported, but that the comparison between species is hard to interpret as presented. I also believe a few analyses should be revisited-not because I expect the qualitative conclusions to change, but because I think they could be refined.

Major comments:

1. As the authors appreciate (e.g., https://peercommunityjournal.org/articles/10.24072/pcjournal.254/), maps inferred from patterns of linkage disequilibrium (LD) are shaped by the ratio of recombination rates to mutation rates, the effective population size, and genome coverage, and therefore cannot be taken at face value. Unfortunately, as a result, the comparison presented in Table 1 is hard to interpret. To do so, the reader would need to know, not just the estimate of rho per bp, but the estimates of Ne (or the estimates of r). It would also be good to provide the estimated ratio of recombination to mutation for each species. 

2. If I understood correctly, the salmon samples were only resequenced to an average of 10X coverage (max 13X). This feature should lead to a substantially worse LD-based map, yet to my knowledge is not discussed, let alone analyzed. I would therefore suggest that the authors add all this information to the main text, and evaluate its possible impact on their conclusions, notably for the degree of overlap in Figure 5C.

3. On a related note, there is less power to detect hotspots in regions of high recombination, so comparisons of hotspot numbers and heats across the genome need to take that into account. As one example, it is not obvious to me that hotspots are actually hotter in telomeres (lines 344-345); perhaps it is rather that only hotter hotspots are detected? 

4. A number of the statistical controls could be tweaked. For example, controls for hotspots are generated without regard to the larger recombination rate/genomic context. Similarly, locations are shuffled without regard to genome gaps, minimal diversity levels etc… It may also be better to use actual genome sequences matched for GC content for the motif searching analyses. To be clear, I am not predicting that the qualitative conclusion will change, but these decisions seem somewhat anti-conservative, and should probably be revisited.

5. This final comment is a matter of opinion, but I thought the framing of the paper was somewhat odd, in making it seem as if we have no idea whether PRDM9 plays the same role outside mammals, when we have evidence that it is active in snakes (cited by the authors) and also evidence that it co-evolves with ZCWPW1 across vertebrates (Cavassim et al. 2022 PNAS; cited elsewhere). In my view, that takes nothing away from the importance of demonstrating that it also directs recombination in fish, especially as this paper presents experimental evidence. In fact, fish are particularly interesting, given the presence of every version of the PRDM9 ortholog within a single taxon. Regardless, I think the authors should report whether ZC1PW1 is present and intact across the four fish species studied here (or only the three with PRDM9), and any differences in their PRDM9 SET domain.

Additional questions/comments:

1. In the phylogenetic analysis, how do the authors convince themselves that the calls are reliable, for example that the copies of alpha 1.2 are actually different?

2. In the experimental analysis, two salmon (TAC1 and TAC3) that share no PRDM9 alleles have 55 hotspots in common, whereas RT52 and TAC3, which share an allele, have only 42. The authors argue that the 55 are likely not real, but are the 42, e.g., do they show the expected asymmetry of reads? 

3. It was unclear to me why the authors chose to use the older version of the motif prediction; do they think it is more reliable? I also wondered if the second motif reported in Figure 3C (which is effectively a series of As) could simply be a nucleosome-excluding sequence.

4. The authors write that "PRDM9 can suppress the recombination activity at chromatin accessible regions (30). Here, we found that this function is conserved in salmonids" (line 562). But unless I missed something, there is no evidence provided for active suppression by PRDM9 in fish-only that rates are not elevated in such regions (e.g., Figure S8 panel C). In that regard, I think it might be worth mentioning in the main text, especially for readers who are not experimentalists, that by looking at DMC1, the data do not strictly reflect double strand frequencies but also potential differences in repair efficiencies.

5. Why does the species without an intact PRDM9 have much larger hotspots (Figure S15) and less of an increase of recombination rates at the telomeres (Figure S16)? Are these real effects? 

6. In the SOM on p. 16, there is what struck me as a surprising claim: that GC rich CpG islands result from GC-biased gene conversion. What is the evidence for this? How does it explain CpG islands in mammals with PRDM9, which do not recombine much in such genomic locations?

Reviewer #2:

I have reviewed the manuscript "PBIOLOGY-D-24-00615_R1" titled "PRDM9 drives the location and rapid evolution of recombination hotspots in salmonids by Marie Raynaud et al." 

The authors present a comprehensive examination of the evolution and function of PRDM9 across Salmonids. The findings imply that PRDM9 function and its mode of action has been conserved for at least several hundred million years. Salmonid recombination shows several unifying features to the well-characterized mammalian models of PRDM9-mediated recombination - despite the fascinating genome duplication event in their past. Dependent on the presence of a full-length Prdm9 genotype, there was evidence for rapid evolution of the PRDM9 gene and positive selection on particular amino acids responsible for DNA binding. Furthermore, Meiotic DNA double-strand breaks (DSBs) maps (a proxy for recombination hotspots) concentrated outside promoter regions and displayed abundant H3K4me3 and H3K4me36. Finally, a fast turnover of recombination hotspots brought on by PRDM9 target motif erosion was shown by population-scaled recombination maps. Overall, this is a very nice paper presenting several significant findings. It is the first evidence of PRDM9-mediated recombination features in fishes. It provides an important piece in the puzzle of how conserved the features of PRDM9-mediated recombination are across much longer evolutionary timescales than previously evaluated. 

I recommend publication upon minor revision. 

Mayor: 

Please clarify: 

For the motif-enrichment analyses, you compared real hotspot data, with a control set of random sequences of equal size and GC-content, and I believe I understand how you generated these random sequences. You state (in the methods), "We retrieved a number of random windows equal to the number of hotspots with a similar GC content distribution"… For me to understand the control dataset, could you state the GC content of the real and random sequences and how "similar" it is to the real dataset? Can you perhaps state what the margins are (in % difference)? Also, did you check whether these "random" loci overlap any of the true hotspots? I believe this information to be crucial. 

Graphical representation of the Supplementary material:

While, in general, the quality of the graphical representations was high in the main body of the manuscript, I was disappointed with the quality of some of the Supplementary Figures. Even though they display exciting findings, they miss the level of detail awarded to the main body of the text and seem to be rather poorly put together. There are several minor and not-so-minor problems that I have listed below:

In Figures S10 and S11, Legend and Title overlap, making the legend hard to read. 

S15, A, and S16 in A: could the number of hotspots be brought on the same scale? Ie. For S15 ~15000 for both OKIS and OMYC and 4000 for the three Salmo salar, 400 for DLAB and the differences explained somehow? It's curious why the "outgroups" to Salmo salar have more than 3-fold difference in number of hotspots. And for S16, a maximum of 250 hotspots on the y-axis would fit all the data. 

S17: "comparaison" should probably read "comparison", and "expectated" should probably read "expected."

Figure S18: comparisons between species are on different scales. i.e. O. mykiss, S. salar NS, BS and GP all have the max value of 4000 shown on the heatmap scale, and O kisutch has 3000. So, the heatmap is hard for me to interpret when all species' graphs for each variable (GC content, SNP density, etc.) are not on the same heatmap scale. I would find it helpful if the X-axis were on the same scale, i.e. -6 to 0 for the Recombination rate. In a perfect world, Y-axes would also match, but I find it understandable that the Y-axis has different scaling. Similarly, I have a hard time interpreting whether the TE density is different in O mykiss and O kisutch compared to the S. salar species, or whether the difference of the shape is driven by plotting at a different scale. In the same context, it would be nice if the Spearman rho values were all on the same horizontal line 

Figure S19, at least SNP density could be put on the same scale, and all plots could have the same height; the legend for the S. salar populations would still fit if the plots were the same height. S23 lacks the top ruler number (3000) The same is seen in Figure S25, where the Frequency of PRDM91 and PRDM9 motifs are shown on different scales, S24, again, the n_neighbors heatmap is on various scales (0-6500) in one, (0-6000) in the other. 

In Figure S26, the probability values are on different scales, from 0.000000 to 0.000100 (six decimals) in one graph and go up to 0.00025 (shown in 5 decimals) in another. The spacing and the numbers on the ruler also differ between graphs. This is again making comparisons extremely difficult to grasp visually. Please unify as much as possible. The p-value also overlaps with the X-axis, making it hard to read. 

Minor:

Line 37: … in turns… Do you perhaps mean "in turn"? This phrase could also be deleted as such: "It increases genetic diversity by creating novel allele combinations (4, 5) that facilitate adaptation and the removal of deleterious mutations from natural populations (6-8)."

Line 41: "Broad-scale patterns of variation within chromosomes (megabase scale) have…"

Perhaps consider rephrasing to "…(at the megabase scale)... 

Lines: 50 - 55: how about in canids? Also, the sentence starting in line 52, "In vertebrates…" is misleading, as only "default" "recombination hotspots are associated with TSSs that are located within CpG islands" in some vertebrates. But this is not generally true for all vertebrates, as you also pointed out later. Please rewrite this sentence to clarify the fact. 

Line 100: "here" may be unnecessary in this context; to me the sentence makes more sense as 

"To this aim, we investigated…"

Line 574 …". This telomere-proximal effect appears to be a conserved property, but of variable strength between sexes and among species with/without Prdm9"

I think it's important to note that telomere proximal enrichment of recombination hotspots is seen in humans and has been seen in dogs and birds as well (that lack PRDM9), including wild birds.

Line 609: "… suggesting that PRDM9 may be limiting in some contexts." 

Do you mean that "PRDM9 dosage" may be limiting? Please clarify

977: I believe the name is Rajalekshmi Navaryana Sarma, not "Sarna."

Reviewer #3:

The work by Raynaud et al., is a straightforward investigation on the determinants of meiotic recombination in salmonids. They are a diverse family of teleost fish, where, crucially, a full length copy of the protein PRDM9 has been found. PRDM9 controls the location of recombination hotspots in mammals, though it has been recently shown that it can direct recombination in vertebrates. PRDM9 has a fascinating history of losses through evolution therefore it is of interest to characterize the recombination landscape in other non-mammalian taxa. Raynaud et al., analyzed the evolutionary dynamics of PRDM9 across the phylogeny of salmonid, identified the sites of recombination initiation in the rainbow trout and inferred historical recombination rates from patterns of linkage disequilibrium (LD) in five populations from three different species. The authors convincingly show that PRDM9 controls recombination hotspot activity in salmonids and that it is the predominant pathway utilized in these fish. Interestingly they also describe a PRDM9-independent pattern of elevated double-strand break formation and recombination in the telomere-proximal region, similarly to what was described in mammals. 

Altogether, this is a very comprehensive and well-executed set of experiments that fully support the major conclusions of the paper. I recommend its publication without reservations as it provides an important piece in the puzzle of how recombination arose and is controlled in and beyond mammals.

A minor comment:

The PRDM9 binding motif derives is not GC-rich as it is in mammals. The authors also show that the GC content is not elevated at recombination hotspots (in a fine-scale). Any thoughts on how these things can be related, and whether GC-biased gene conversion is not as active because of the genomic context and not due to some intrinsic mechanistic differences?

Reviewer #4:

In this manuscript entitled "PRDM9 drives the location and rapid evolution of recombination hotspots in salmonids", Raynaud et al. analyzed the meiotic recombination in different species of salmonids with a focus on the impact of Prdm9 allelic background. The authors used fine-scale recombination maps based on DMC1 ChIP sequencing and estimation of population-scaled recombination rates to present descriptive data of several salmonids. This is combined with an in silico analysis of the genomic features associated with hotspots. Authors conclude that PRDM9 determines location of recombination hotspots, leading to evolutionary signatures on those sites such as rapid motif erosion and turnover of recombination hotspots, resembling previous described patterns in model mammalian species. 

Authors provide a nice introduction with relevant background material and the results are clearly written with well-organized sections, making it easy to interpret. Overall, results presented are compelling, being of broad interest to the meiosis community, especially for those working in non-model species. However, conclusions are not entirely novel, beyond the description of a conservation of patterns already described in mammals. Perhaps it will be desirable to make more emphasis on those patterns that are not common among mammals (such as mice or humans). In relation to that, authors might extend their interpretations on the data of truncated Prdm9 paralogs, providing additional detail on the possible implications.

Major comments

1. All throughout the work (see lines 255-265 or 340), authors compare their results on salmonids species with mice/humans recombination hotspots. However, the number of peaks in SSDS are considerably low compared to those detected in mice or humans. This needs clarification. Would it be related to a limitation of the technique when used in salmonids, including the use of different antibodies for different animals/species? Also, the number of detected hotspots based on LD are much higher. How can the authors reconcile these contrasting results? 

2. Related to the previous point, pertinent comparisons with patterns found in non-mammalian species (such as snakes or birds) are missing. Also, there is not any mention of studies in meiotic recombination from other fishes.

3. Limited reference is made to the differences in intensity of hotspots, particularly on those shared or specific per alleles (they compare intensities when comparing hotspots from SSDS and LD at line 354, or in the discussion-Line 575-, but not between the studied species/populations) 

4. Discussion on the genomic features characterizing independent-prdm9 hotspots is particularly absent.

5. It would be interesting to pointing out genomic features of independent-prdm9 hotspots. It was especially absent in the discussion.

6. Main figures could benefit from a clearer focus to strength the manuscript. For example, figures can be more homogeneous when considering aesthetics; figure 4 seems differently produced in terms of font size and style than the rest. 

Additional comments:

7. Only GC-biased gene conversion is not a common pattern between salmonids and mammals-birds (Lines 405-409), is that right?

8. Regarding the recombination of tetraploid genomic regions is commented on the LD data. Have they compared the number of recombination hotspots in SSDS derived data? (Lines ~422-435).

9. Line 467: Authors attribute the high percentage of shared hotspots to common Prdm9 alleles, but they should also indicate the similar genetic background.

10. Since they have the prdm9 sequences of the alleles, looking for the predicted motif to compare will be informative. Reported motifs show strange pattern given previous literature, especially the second has high percentage of T. Aren't motifs enriched in GC?

11. Authors do not show examples of histological sections from testis, this will be useful to interpret meiosis progression in the samples used.

12. As for the genome assemblies used, it will be informative to provide stats on genome continuity; how many scaffolds and how large; are all scaffolds assigned to chromosomes?

Minor points:

* Line 72: Please, specify which Mus subspecies; there are several works describing differences between them.

* Line 178: Ts3R event (Ss4R)?

* Line 205: Please, include the reference to 'other species'.

* Lines 285-295: This section is hard to follow.

* Line 354: four out of five salmonids… which ones?

* Data support statement of Lines 423-425 needs to be included.

* Explain better the percentages of Lines 410 and 411.

* Population abbreviations are not indicated until methods.

* Line 458-459: Please, include number of hotspots, not just percentages.

* Line 488: Missing reference.

* Line 560: Perhaps will be more informative to include relevant references of recombination analysis in wild mice having prdm9 allelic variation.

* Line 623: it is stated that …"may be even older", but in the intro (Line X) they already citate papers that indicate that prdm9 was ancestral to all animals.

* Line 641: "a recent study in mammals…", please indicate the reference.

* Line 699: Is paralog alfa2.2 located in different chromosomes for S Salar and O mykiss? Chr 7 and 17?

---

## [Decision Letter · Decision Letter 2]

25 Oct 2024

Dear Dr de Massy,

Thank you for your patience while we considered your revised manuscript "PRDM9 drives the location and rapid evolution of recombination hotspots in salmonids" for publication as a Research Article at PLOS Biology. This revised version of your manuscript has been evaluated by the PLOS Biology editors, the Academic Editor and three of the original reviewers.

Based on the reviews , we are likely to accept this manuscript for publication, provided you satisfactorily address the remaining points raised by the reviewers and the following data and other policy-related requests.

IMPORTANT - please attend to the following:

a) Please make the Title slightly more explicit for our broader readership: "PRDM9 drives the location and rapid evolution of recombination hotspots in salmonid fish"

b) Please address the remaining minor concerns from reviewer #4.

c) I note that you use new fish samples for PCR and ChiP-seq ("We used wild Atlantic salmon samples from Normandy (France) and rainbow trout samples from an INRAE selected strain"), so please can you clarify whether you needed ethical approval and/ or field licences for these samples?

d) Please address my Data Policy requests below; specifically, we need you to supply the numerical values underlying Figs 1A (treefile), 2BC, 3ABCD, 4ABC, 5ABCD, S1 (treefile), S5BC, S6ABC, S7AB, S8ABC, S9AB, S10AB, S11, S12, S13, S14ABD, S15ABC, S16AB, S17, S18AB, S19AB, S20ABCD, S21ABC, S22, S23, S24ABC, S25AB, S28, S29ABCDEF, S30ABCDEFGHI, either as a supplementary data file or as a permanent DOI’d deposition. I note that you already have an associated Zenodo deposition (https://doi.org/10.5281/zenodo.11083953), but please could you clarify whether this contains all of the data and code needed to recreate the Figures?

e) Please cite the location of the data clearly in all relevant main and supplementary Figure legends, e.g. “The data underlying this Figure can be found in S1 Data” or “The data underlying this Figure can be found in https://zenodo.org/records/11083953"

f) Please make any custom code available, either as a supplementary file or as part of your Zenodo deposition.

We expect to receive your revised manuscript within two weeks. 

*Published Peer Review History*

*Press*

Sincerely,

Roli Roberts

Roland Roberts, PhD

Senior Editor

rroberts@plos.org

PLOS Biology

ETHICS STATEMENT:

-- Please include the full name of the IACUC/ethics committee that reviewed and approved the animal care and use protocol/permit/project license. Please also include an approval number.

-- Please include the specific national or international regulations/guidelines to which your animal care and use protocol adhered. Please note that institutional or accreditation organization guidelines (such as AAALAC) do not meet this requirement.

DATA POLICY:

Regardless of the method selected, please ensure that you provide the individual numerical values that underlie the summary data displayed in the following figure panels as they are essential for readers to assess your analysis and to reproduce it: Figs 1A (treefile), 2BC, 3ABCD, 4ABC, 5ABCD, S1 (treefile), S5BC, S6ABC, S7AB, S8ABC, S9AB, S10AB, S11, S12, S13, S14ABD, S15ABC, S16AB, S17, S18AB, S19AB, S20ABCD, S21ABC, S22, S23, S24ABC, S25AB, S28, S29ABCDEF, S30ABCDEFGHI. NOTE: the numerical data provided should include all replicates AND the way in which the plotted mean and errors were derived (it should not present only the mean/average values).

CODE POLICY

DATA NOT SHOWN?

REVIEWERS' COMMENTS:

Reviewer #1:

The authors have responded to all my concerns and suggestions.

Reviewer #2:

[identifies herself as Dr. Linda Odenthal-Hesse]

I have reviewed the revised manuscript "PBIOLOGY-D-24-00615_R2" titled "PRDM9 drives the location and rapid evolution of recombination hotspots in salmonids by Marie Raynaud et al."

The authors have addressed my previous concerns and recommendations with great care, and I am fully satisfied with the revisions. I also feel that the comments made by other reviewers were carefully addressed. This is a well constructed paper and I recommend publication.

Reviewer #4:

The authors have satisfactorily addressed my initial comments. The manuscript is now much clearer, with more coherent argumentation and a stronger effort to discern the differences between species. I believe this paper makes a significant contribution to the field, and with a few minor revisions, it will be ready for publication. Minor comments.

* Line 257, the percentage doesn't should be notified in % rather than decimals?

* The sequence of the identified prdm9 alleles will be reported in fasta format? I can see in Zenodo some processed datasets such as VCF files, LD-based recombination maps and hotspots, PRDM9 protein sequences and multiple alignments, but not allelic variants

* Line 426, "On average, 90% of the total recombination appeared to be concentrated in 20% of the genome, a higher rate than what was observed in human and chimpanzee (29, 72) and slightly higher than what we observed in sea bass (Table 1, S12 Fig, S2 Table). This heterogeneity was largely driven by the presence of recombination hotspot" - From the text, I understand that the differences between sea bass and other salmonids are related to the presence of recombination hotspots and PRDM9. However, how do they explain why the recombination rate is higher in sea bass than in humans and chimpanzees?

* Line 418-419, there is an inconsistency between the recombination rates in the text and in the table 1 (either 0.012 or 0.0012)

* Figures are difficult to see, likely caused by the PDF format. Specifically, Figure 1A, which contains a lot of information and species names, is hard to read. Additionally, Figure 4, panels B and C, seem confusing. The blue and purple colors of the TSS in and out CGI bars are uniquely colored, but their legend is missing in the panels. Overall, the information and main point of Figure 4 is hard to interpret.

---

## [Editor Report · Decision Letter 3]

21 Nov 2024

Dear Bernard,

Thank you for the submission of your revised Research Article "PRDM9 drives the location and rapid evolution of recombination hotspots in salmonid fish" for publication in PLOS Biology. On behalf of my colleagues and the Academic Editor, Nick Barton, I'm pleased to say that we can in principle accept your manuscript for publication, provided you address any remaining formatting and reporting issues. These will be detailed in an email you should receive within 2-3 business days from our colleagues in the journal operations team; no action is required from you until then. Please note that we will not be able to formally accept your manuscript and schedule it for publication until you have completed any requested changes.

Sincerely, 

Roli

Senior Editor

PLOS Biology

rroberts@plos.org